# PRIVET: PRIVACY METRIC BASED ON EXTREME VALUE THEORY

## ABSTRACT

Deep generative models are often trained on sensitive data, such as genetic sequences, health data, or more broadly, any copyrighted, licensed or protected content. This raises critical concerns around privacy-preserving synthetic data, and more specifically around privacy leakage, an issue closely tied to overfitting. Existing methods almost exclusively rely on global criteria to estimate the risk of privacy failure associated to a model, offering only quantitative non interpretable insights. The absence of rigorous evaluation methods for data privacy at the sample-level may hinder the practical deployment of synthetic data in real-world applications. Using extreme value statistics on nearest-neighbor distances, we propose PRIVET, a generic sample-based, modality-agnostic algorithm that assigns an individual privacy leak score to each synthetic sample. We empirically demonstrate that PRIVET reliably detects instances of memorization and privacy leakage across diverse data modalities, including settings with very high dimensionality, limited sample sizes such as genetic data and even under underfitting regimes. We compare our method to existing approaches under controlled settings and show its advantage in providing both dataset level and sample level assessments through qualitative and quantitative outputs. Additionally, our analysis reveals limitations in existing computer vision embeddings to yield perceptually meaningful distances when identifying near-duplicate samples.

## 1 INTRODUCTION

### 1.1 POSITION OF THE PROBLEM

In the context of unsupervised learning, recent advances in deep learning have enabled the development of generative models capable of producing highly realistic artificial data from a collection of examples. Prominent examples include Generative Adversarial Networks (GANs)(Goodfellow et al., 2014), Variational Autoencoders (VAEs) (Kingma & Welling, 2014), normalizing flows Rezende & Mohamed (2015), diffusion models Ho et al. (2020), flow matching techniques Liu et al. (2022), energy-based models LeCun et al. (2006) or Large Language Models (LLMs) Vaswani et al. (2017). When applied to image data, these models can synthesize remarkably lifelike samples – for instance, photorealistic human faces that do not correspond to real individuals. One major difficulty is to assess the quality of these models. A widely used metric is the Fréchet Inception Distance (FID) score Heusel et al. (2017), allows one to assess the visual quality and diversity of the generated samples, but is not specifically designed to detect overfitting or privacy leakage and often fails to do so Jiralerspong et al. (2023). Despite their high capacity to imitate real data distributions, generative models are prone to overfitting due to their typically large number of trainable parameters, particularly when trained on limited datasets. This vulnerability affects many applications and becomes particularly critical in domains such as genomics, where the training data may contain identifiable and sensible information – for example, genetic sequences associated with specific diseases. As discussed in Yeom et al. (2018), privacy leakage and overfitting are closely related phenomena, yet conceptually distinct. Overfitting may facilitate privacy attacks, but privacy leaks can occur without it. Multiple forms of privacy leakage have been identified in

the literature, the most commonly studied being *membership inference* and *attribute inference*. Membership inference aims to determine whether a specific sample was used during training, while attribute inference seeks to reconstruct sensitive attributes of individuals from model outputs. Additional privacy threats include model inversion and influence-based attacks, applicable under varying levels of adversarial access. These attacks relate to broader notions of stability in machine learning (ML), and raise serious concerns in domains involving sensitive data, such as genomics. Privacy leaks in genetic data were identified early on Homer et al. (2008); Sankararaman et al. (2009), with recent work introducing formal metrics and empirical evaluations Oprisanu et al. (2021). Simmons & Berger (2015) developed a way to detect leakage at the sample level by only looking at summary statistics of genetic variation data. However, summary statistics rely on handcrafted transformations of the data, often involving expert knowledge; hence, such approaches are domain-specific. Instead, we want to develop a generic metric that is applicable across data domains. While our focus is on privacy risks in generative models, we acknowledge the broader privacy challenges in ML ( see Del Grosso et al. (2023) for a wider overview).

## 1.2 RELATED WORK

Evaluating synthetic data is a critical step that extends beyond the development of generative models. Such data must satisfy key desiderata, including high quality – capturing both fidelity and diversity – and *novelty*, i.e. no privacy leakage or memorization (i.e. regurgitation of training samples).

Among existing evaluation metrics, the holistic approaches aim to broadly capture all these aspects simultaneously. For example, Jiralerspong et al. (2023) introduced a score that correlates with fidelity and diversity, and can also be adapted to assess memorization and overfitting. In contrast, property-specific metrics focus on evaluating a particular aspect of the data. Our proposed method aligns with that, offering a granular assessment of proximity-based privacy leakage at the sample level, which can further be extended to a global measure.

**Metrics at the dataset-level.** In the context of sensitive data, Yale et al. (2020a) introduced the Nearest Neighbor Adversarial Accuracy ($\mathcal{AATS}$), a score in $[0, 1]$ estimating the resemblance between generated and real training data. A value below $0.5$ indicates overfitting, while above $0.5$ suggests underfitting. This score can also be adapted into a privacy metric by comparing its value to that obtained from a separate test set. $\mathcal{AATS}$ requires equal sample sizes between training and synthetic data, which can be limiting, especially when test sets are significantly smaller.

In that vein, $C_T$ Meehan et al. (2020) is a statistical test to detect anomalous resemblance of generated data to training ones. More recently, Jiralerspong et al. (2023); Lemos et al. (2025b) proposed holistic approaches, Feature Likelihood Divergence (FLD) and PQMass, that rely on kernel density estimation and statistical tests on counts of data points falling in cells constructed by a tesselation of space. These approaches have important limitations: they fail in low-data regimes or produce very noisy estimates, and lack interpretability, i.e., they output continuous scores that neither identify which samples are leaked nor convey how many individuals are

Table 1: **Privacy Metrics:** Overview of interpretability across existing privacy evaluation metrics. Metrics shown in blue background are **interpretable**, whereas those in purple are **not interpretable**.

| Papers | Dataset-level (global) | Sample-level (local) | |
|---|---|---|---|
| | | **Real value** | **Boolean (leak or not)** |
| Yale et al. (2020b) ($\mathcal{AA}_{\mathcal{TS}}$) | ✓ | ✗ | ✗ |
| Meehan et al. (2020) ($C_T$) | ✓ | ✗ | ✗ |
| Alaa et al. (2022) ($Auth$) | ✓ | ✗ | ✓ |
| Jiralerspong et al. (2023) ($FLD$) | ✓ | ✓ | ✗ |
| Lemos et al. (2025b) ($PQM$) | ✓ | ✗ | ✗ |
| Ours ($PRIVET$) | ✓ | ✓ | ✓ |

affected. Such shortcomings hinder their use in downstream tasks like sample selection, despite the need for precise, transparent metrics in privacy evaluation—an issue underscored by regulations such as the GDPR gdp (2016).

**Metrics at the sample-level.** Like $\mathcal{AATS}$, the Authenticity score Alaa et al. (2022) uses nearest-neighbor distances and provides an interpretable "leak" or "no-leak" label for each synthetic sample, but leaves aside statistical aspects, making it vulnerable to fluctuations where a synthetic sample may seem close to a training point by chance. FLD can additionally be used to assign a real number (cf. definition 3.1 in Jiralerspong et al. (2023)) at the sample level, but it still requires manual inspection to confirm privacy breaches.

### 1.3 MAIN CONTRIBUTIONS

In this work, we propose PRIVET, a novel sample-based global and local privacy metric based on nearest-neighbor (NN) distribution statistics at its core, as illustrated on Fig. 1. NN distances are by definition, extreme value random variables. By modeling the distribution of NN distances using extreme value theory, specifically, Weibull or Gumbel distributions, we identify specific synthetic samples that are anomalously close to individual training instances. Our method has several advantages over existing methods. Specifically:

- **Sample-level assessment.** Privacy metrics at the sample level have largely been overlooked in the literature. PRIVET assigns a privacy leak score for each sample in a dataset, hence detecting memorization, overfitting and privacy leakage in underfitting regime.

- **Interpretability.** PRIVET scores can be declined in multiple forms among which probabilities that can be thresholded, yielding a boolean "leak" or "not leak" status that is easily interpretable.

- **Scalability.** PRIVET inference is fast, scalable to high dimension and robust to low data regime.

- **Versatility.** PRIVET is domain-agnostic, meaning it can be applied on any type of data.

We tested our method extensively and compared it to existing ones using different datasets from different domains, namely genetics and computer vision, where privacy preservation can be crucial. Furthermore, we considered two types of generative models – RBMs (as prototypical energy-based models), and diffusion models – to demonstrate the applicability and robustness of our privacy assessment framework across diverse generative paradigms.

## 2 THE METHOD

### 2.1 BASIC ASSUMPTIONS AND HYPOTHESIS ON NEAREST-NEIGHBOR DISTANCES

The working hypothesis is that overfitting and privacy leaks that manifest as proximity-based anomalies are reflected in distortions of nearest-neighbor distances between synthetic and train or test samples. Then the main point underlying the method is that assessing whether a generated sample is too close from a train sample is a statistical question which can be dealt with *extreme value statistics*.

Assume we have a collection $\mathcal{T}_N = \{\boldsymbol{x}_i, i = 1, \ldots N\}$ of $N$ examples, drawn independently from a probability distribution with density $\rho$. The points $\boldsymbol{x}_i$ are embedded into a feature space which can be discrete or continuous. In addition we assume that we are given a meaningful distance $d(\boldsymbol{x}, \boldsymbol{y})$ on this space. For genetic sequences for instance it makes sense to use the Hamming distance.

Given a point $\boldsymbol{x}$ sampled from the density $\rho$, let $\delta(\boldsymbol{x}, N)$ represent the *nearest-neighbor* (NN) distance of $\boldsymbol{x}$ to the set $\mathcal{T}_N$ of examples, all sampled as well from $\rho$ :

$$\forall \boldsymbol{x} \sim \rho, \quad \delta(\boldsymbol{x}, N) \stackrel{\text{def}}{=} d(\boldsymbol{x}, \boldsymbol{y}) \quad \text{where} \quad \boldsymbol{y} = \arg\min_{\boldsymbol{y}' \in \mathcal{T}_N} \Big[ d(\boldsymbol{x}, \boldsymbol{y}') \Big].$$

Then we denote by

$$F(u, N) = \mathbb{E}_{\boldsymbol{x} \sim \rho} \big[ P\left( \delta(\boldsymbol{x}, N) < u \right) \big]$$

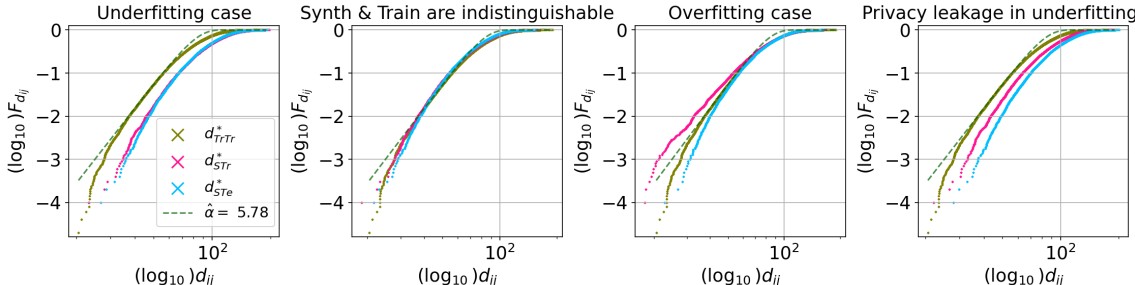

Figure 1: **Schematic figure.** Empirical cumulative distribution functions (eCDFs) of nearest-neighbor distances allow for distinction of generative scenarios. $d^*_{TrTr}$, $d^*_{STr}$, and $d^*_{STe}$ denote the distributions of nearest-neighbor distances from train to train, synthetic to train, and synthetic to test, respectively. We plot the eCDF for each of these distributions and fit the tail of $d^*_{TrTr}$ with an extreme value distribution either Weibull or Gumbel (dark green dashed line). The x- and y-axes represent the distances and the eCDF, respectively, and are both shown on a $\log_{10}$ scale. When eCDF of $d^*_{STr}$ and $d^*_{STe}$ are below eCDF of $d^*_{TrTr}$, ie synthetic to train and synthetic to test nearest-neighbor distances are higher than train to train nearest-neighbor distances, we are in the underfitting case. When all eCDFs are aligned, synthetic are indistinguishable from training and test samples. When eCDF of $d^*_{STr}$ is inflated with respect to $d^*_{TrTr}$ and $d^*_{STe}$, ie distances in $d^*_{STr}$ are lower, then we are in the overfitting case. Privacy leaks are detected by measuring the deviation between the $d^*_{STr}$ and $d^*_{STe}$ which may occur both in the overfitting and underfitting regime.

the probability that this random variable is at a distance smaller than $u$. For large $N$, we expect the distribution to exhibit a universal behavior in the small-$u$ regime in the form of an extreme value distribution (EVD) Haan & Ferreira (2006). To express it, let us consider the pairwise distance distribution between pairs of examples sampled from $\rho$ :

$$F(u) = \int d^D\boldsymbol{x}\, d^D\boldsymbol{y}\, \rho(\boldsymbol{x})\rho(\boldsymbol{y})\mathbf{1}_{\{d(\boldsymbol{x},\boldsymbol{y})<u\}}$$

where $d^D$ refers to the infinitesimal volume element in $D$-dimensional space. From the fact that the NN of $\boldsymbol{x}$ in $\mathcal{T}_N$ is the result of $N$ independent draws of $\boldsymbol{y} \sim \rho$ , we have

$$F(u, N) \leqslant 1 - \big[1 - F(u)\big]^N,$$

where the inequality becomes an equality when the density $\rho$ is uniform. This upper bound becomes a universal limiting distribution for large $N$, determined by the asymptotic behavior of $F(u)$ as $u \to 0$. In turn this will be dominated by regions of large density for small $u$, which yields a lower bound for $F(u, N)$, so the lower tail of $F(u, N)$ is expected to coincide with the tail of its upper bound, which justifies to use an EVD fit (more details in supplementary material).

## 2.2 EXTREME VALUE DISTRIBUTION FOR NEAREST-NEIGHBOR DISTANCES

When considering NN distances we probe the tail of the pairwise distances distribution. In that case we are in the regime of extreme value statistics which displays some universal behavior. Extreme value theory (see e.g. Haan & Ferreira (2006)) asserts under some conditions the existence of two sequences $a_N$ and $b_N$ s.t. the rescaled variable $v = (u + b_N)/a_N$ has a limit probability distribution when $N \to \infty$, i.e.

$$\lim_{N\to\infty}\big[1 - F(a_N\,v - b_N)\big]^N = G(v),$$

with $G$ a non-degenerate distribution function representing the probability that the rescaled minimum distance $[\delta(\boldsymbol{x}, N) + b_N]/a_N$ is greater than $v$. The shift $b_N$ tells how $u$ should scale with $N$ while $a_N$ controls the level of fluctuations near this trend. Depending on the tail (either power law or Gaussian) $G$ corresponds to a

Weibull or Gumbel distribution (see Supp.Mat. for details). In practice the fit is performed for a given $N$ with maximum likelihood estimation both on the Weibull and Gumbel cases, and we select the one with highest likelihood. In both cases we have two parameters to determine:

$$\hat{G}(u) = \begin{cases} e^{-\hat{A}\,u^{\alpha}} & \text{(Weibull),} \\ e^{-\hat{A}\,e^{\hat{B}\,u}} & \text{(Gumbel),} \end{cases} \tag{1}$$

where $u$ is the non-rescaled distance, the rescaling being implicitly contained in $\hat{A}$ and $\hat{B}$, keeping in mind that these fitting coefficients depend on the size $N$ of the dataset against which the NN search is defined and have to be properly rescaled when e.g. using train and test sets of different sizes. Quality of the fits to the NN distributions was assessed via goodness-of-fit tests and showed close agreement with the EVT hypothesis across datasets and modalities (cf. Appendix D).

Consider now the situation where we generate $M$ synthetic examples $\{\boldsymbol{x}_i, i = 1, \dots M\} = \mathcal{S}_M$. To each $\boldsymbol{x}$ of these we associate a NN distance $\delta(\boldsymbol{x}, N)$ to the set $\mathcal{T}_N$, which is now drawn from the extreme value probability distribution $G$. In principle these random variables are not independent because $\mathcal{T}_N$ is fixed, therefore we expect a residual dependency for close-by synthetic examples. We leave this aside and assume independence as we expect these dependencies to have sub-dominant effects on the analysis. Given the fit $\hat{G}$, the probability to find among $M$ NN distances the $r$th smallest one noted $\delta_r(N, M)$ at a value smaller than $u$ is a binomial estimated from the fit as:

$$P\big[\delta_r(N, M) < u\big] \simeq 1 - \sum_{q=0}^{r-1} \frac{M!}{q!(M-q)!} \hat{G}(u)^q \big[1 - \hat{G}(u)\big]^{M-q}.$$

Let us call this key quantity $P_{N,M}(u, r)$. This $r$th order statistics represents the probability to actually observe such rank $r$ event whenever the synthetic data are undistinguishable from the real ones.

### 2.3 SCORES BASED ON $r$TH ORDER STATISTICS

At this point we have everything needed to compute various overfitting and privacy metrics of interest. First we introduce an individual overfitting score $\pi_r^{\text{ref}}$ associated to the sample $\boldsymbol{x}_r$ extracted from a dataset of interest, relatively to a reference dataset, typically the train or test set, indexed by increasing order w.r.t. NN distances. This score is given by:

$$\pi_r^{\text{ref}} = P_{N,M}(u_r, r)$$

where $N$ is the size of the reference dataset, $M$ the size of the dataset of interest, $u_r$ the $r$-th smallest NN distance to the reference dataset, among the $M$ values computed from the points of the dataset of interest. A low value of this score indicates that $u_r$ is too small to be at the rank $r$ due to overfitting. As privacy leaks may occur also in the underfitting regime Yeom et al. (2018), to detect them we need to take a reference probability based on the test set. This leads us to introduce the individual privacy score:

$$\Delta\pi_r \overset{\text{def}}{=} \log_{10}\left(\frac{\pi_r^{\text{train}}}{\pi_r^{\text{test}}}\right). \tag{2}$$

In other words, for a given synthetic sample, we compute the log ratio of overfitting scores with respect to the training and test sets of real data. A low value indicates a statistical anomaly and suggests a potential privacy leak. Unlike standard overfitting, a low privacy score can arise even if the training probability $\pi_i^{\text{train}}$ is not particularly small. This leads naturally to the definition of two global privacy leakage indices.

$$\Delta\pi = \frac{1}{M} \sum_{r=1}^{M} \Delta\pi_r, \qquad \text{NPL} = \sum_{r=1}^{M} \mathbf{1}_{\{\Delta\pi_r < \tau\}}. \tag{3}$$

The first one indicates the average level of privacy leakage, the second one being the estimated number of synthetic samples breaking privacy given the threshold $\tau$. Typically we will set $\tau = -3$. A more refined

quantitative estimation of the number of overfitting samples or privacy leaks can be obtained as follows. For a given distance $u$ we can obtain this by comparing the actual rank observed for this distance with the rank $r(u)$ corresponding to

$$P_{N,M}(u, r(u)) = \frac{1}{2},$$

the difference between the two giving the excess (or deficit if negative) of samples having NN distance below $u$. For large $M$ this rank is directly read off from $\hat{G}$:

$$r(u) \approx M\hat{G}(u) \qquad (r(u) \gg 1)$$

As a result $r_u - r(u)$ represents the excess of samples having NN distance below $u$ if $r$ is the rank of the sample having NN distance $u$ (more details in Appendix B). This leads to define the following quantities:

$$n_{\text{overfit}}(r) \stackrel{\text{def}}{=} r - r[u_r^{\text{train}}] \tag{4}$$

$$n_{\text{pleaks}}(r) \stackrel{\text{def}}{=} r[u_r^{\text{test}}] - r[u_r^{\text{train}}] \tag{5}$$

where $u_r^{\text{train}}$ and $u_r^{\text{test}}$ represents the $r$th NN distance respectively to the train and to the test dataset. $n_{\text{overfit}}(r)$ provides us directly with an estimation, among the first $r$ synthetic samples of lowest NN distance to the train, of the expected number of overfitting samples; $n_{\text{pleaks}}(r)$, by making use of the NN distance corresponding to the same rank $r$, w.r.t. the test set, actually measures the excess of small NN distances to the train, compared to the test, among the same first samples $r$. We refer the reader to Appendix B for details on how $n_{\text{excess}}(r) \stackrel{\text{def}}{=} r - r[u_r^{\text{ref}}]$, the expected excess of samples up to rank $r$ whatever reference set is considered (either train of test), can be converted into a local privacy leak index (Eqs. 10).

## 3  CONTROLLED EXPERIMENTS

### 3.1  GENETIC DATA

Here we perform a controlled experiment where a certain amount of information from the training data is leaked into the pseudo-synthetic data. These data are real genetic data corresponding to binary sequences of size 65535, retrieved from The 1000 Genomes Project Consortium et al. (2015) (see Appendix E.1 for further details). Distance between samples is simply the Hamming distance. We partition these 5004 samples into three random subsets each of equal size 1668 and arbitrarily assign these to train, test and synthetic datasets, the latter corresponding to the output from a fictional generative model. Then we set two parameters, $f_{\text{fake}} \in [0, 1]$ and $f_{\text{copy}} \in [0, 1]$, corresponding respectively to the fraction of synthetic samples containing leaked information from the training data (i.e., percentage of copied samples) and the fraction of leaked information (i.e., percentage of copied segments). The synthetic data are constructed as follows. In the subset corresponding to synthetic data, a random fraction $f_{\text{fake}}$ is selected at random to contain leaked information. For each of these assigned samples, we first identify its nearest neighbor in the training set and replace in its sequence a fraction $f_{\text{copy}}$ of random bits copied from the training sample, which model the private information assumed to be leaked by the generative model. To demonstrate robustness, we repeated the experiment on a low-sample-size scenario dataset (see Appendix E.1 for details).

With these ground-truth datasets, we conducted a comprehensive evaluation using a diverse set of metrics, that quantify overfitting in generative models (section 1.2 and Table 1). For $\mathcal{AA_{TS}}$, FLD and PQMass, we considered the generalization gap, i.e., $Score(\mathcal{D}_{test}, \mathcal{D}_{synth}) - Score(\mathcal{D}_{train}, \mathcal{D}_{synth})$ where $Score()$ is one of these metrics. $Privacy\ Loss$ (i.e. the generalization gap of $\mathcal{AA_{TS}}$ along with that of PQMass is strictly positive when overfitting is detected, while those of FLD and $C_T$ are strictly negative).

Fig. 2 compares the behavior of the different global scores when both $f_{\text{fake}}$ and $f_{\text{copy}}$ are varied. We observe that $C_T$, the generalization gap of FLD, and PQMass exhibit limited sensitivity to these changes, notably $C_T$ and the generalization gap appear noisy. In the case of PQMass, the observed noise may also stem from its

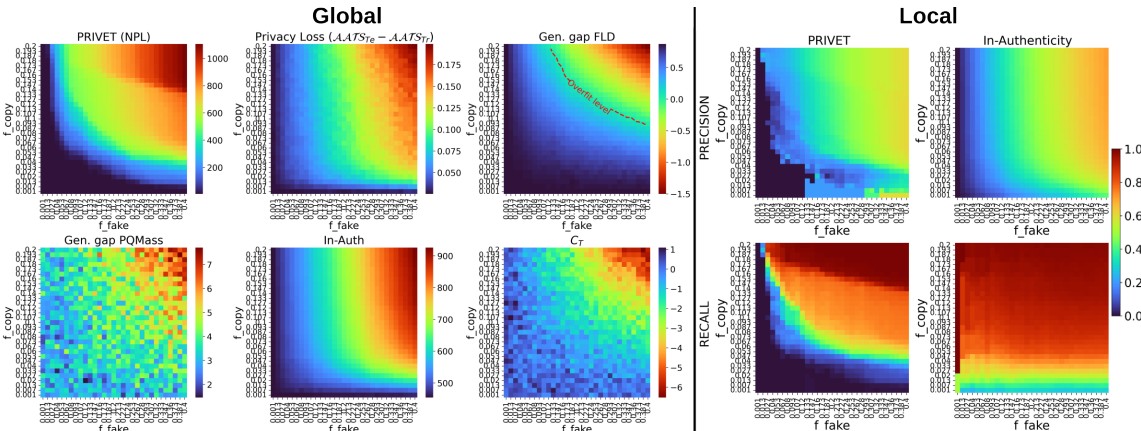

Figure 2: **Global and local privacy scores for 65,535 SNPs and 1,668 samples. Global:** From top left to bottom right—number of privacy leaks (NPL Eqs. 3) from PRIVET, generalization gaps of $\mathcal{AATS}$, FLD, and PQMass, In-Authenticity, and $C_T$, plotted over $f_{\text{fake}}$ vs. $f_{\text{copy}}$. Color indicates metric values; for FLD, the overfit level defines the contour above which overfitting is detected. **Local:** Precision (top) and recall (bottom) are reported for local privacy metrics. PRIVET score $\Delta\pi_r$ (Eqs. 2) was thresholded with $\tau = -3$.

reliance on a single set of reference points used to create the tessellations. A potential solution to mitigate this noise would be performing multiple repetitions varying the reference points (e.g., 20 as performed by the original authors, see Fig. 8 in Lemos et al. (2025a)), but it would be computationally heavy for each $(f_{\text{fake}}, f_{\text{copy}})$ configuration in our case. For gen. gap FLD, the zone above the overfit contour line where a leak signal would be triggered corresponds to already large values of $f_{\text{fake}}$. By contrast, the number of privacy leaks (NPL) detected by PRIVET looks less noisy and more sensitive both to $f_{\text{copy}}$ and $f_{\text{fake}}$. In particular, PRIVET (NPL, Eqs. 3) exhibits a sharp threshold separating a phase where little or no privacy leak is detected (dark domain) at low values of both $f_{\text{fake}}$ and $f_{\text{copy}}$ from the privacy leak phase (colored domain), with as a bonus a precise indication of the number of synthetic data that leaks information. Privacy loss and the negated Authenticity score (In-Auth) also appear highly sensitive to privacy leakage. However, Privacy loss provides only a global (i.e., dataset-level) assessment and is therefore not interpretable, as it does not identify which specific samples are concerned with information leakage. Although In-Auth demonstrates strong precision and recall across both privacy leak axes, PRIVET achieves higher recall for $f_{\text{copy}} \geq 0.15$, as illustrated in Fig. 2. While In-Auth appears overall stronger, it does not account for dataset size and is therefore highly sensitive to statistical fluctuations. As a result, a synthetic sample may appear spuriously close to a training point purely by chance (see FIG. 7). Moreover, by construction, In-Auth cannot detect privacy leakage in the underfitting regime.

## 3.2 IMAGE DATA

PRIVET is a general-purpose algorithm applicable across diverse data modalities, provided the feature representations yield well-behaved distance metrics. To evaluate its versatility, we also benchmarked PRIVET on the copycat experiment Douze et al. (2009); Jiralerspong et al. (2023). We used synthetic data generated by SOTA diffusion model on CIFAR10 Xu et al. (2023) that was provided by Stein et al. (2023).

We applied several transformations on training and synthetic data corresponding to easy and hard distortion scenarios, then created pseudo-synthetic datasets consisting in progressive replacement of transformed synthetic data with transformed training data, and computed embeddings of these data using either DINOv2 Oquab et al. (2024). Finally, we applied the metrics on the embeddings of the non-transformed training set, non-transformed test data and the mixture of transformed synthetic and transformed training data (pseudo-

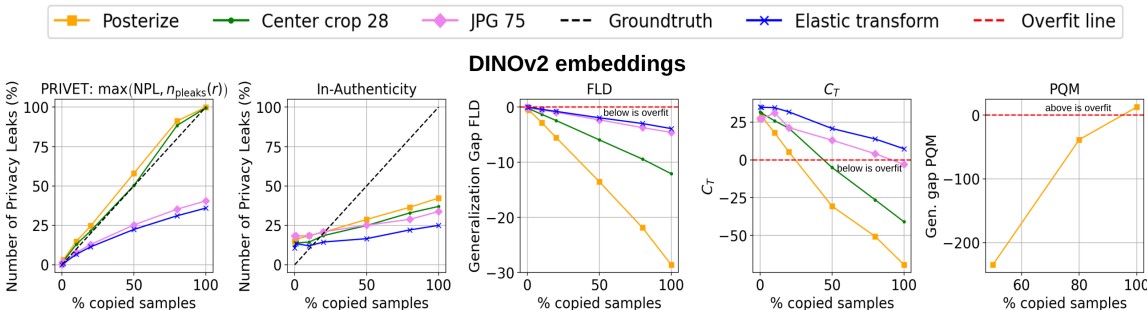

Figure 3: **Copycat experiment.** Several computer vision transformations (posterize, center crop 28, JPG 75, Elastic transform) are applied on CIFAR10 synthetic and training samples to generate a pseudo-synthetic dataset. x-axis represents the fraction of synthetic samples replaced with training ones; y-axis reports the value of the corresponding metric. For PRIVET and In-Authenticity, we report the percentage of identified privacy leaks. In the case of PRIVET, we use a conservative leakage score obtained by taking $\max(\text{NPL}, n_{\text{pleaks}}(r))$. For Gen. gap (FLD) and $C_T$, overfitting is detected when the metric falls below the horizontal dashed line, whereas for Gen. gap (PQMass), overfitting is indicated when the metric exceeds the threshold. DINOv2 (ViT-B/14 distilled without registers) embeddings of CIFAR10 were computed. For several transformation PQMass score diverged to $-\infty$, hence results are omitted.

synthetics). This corresponds to a situation where the generative model produces distorted copies of the training set.

In the case of PRIVET, we use a conservative leakage score defined as $\max(\text{NPL}, n_{\text{pleaks}}(r))$ (see Eqs. 3 & 9). This choice is motivated by the particular behavior induced by DINOv2 embeddings: although embeddings of PFGM++ samples generally appear underfit, the posterize and center crop 28 transformations produce leakage patterns in which some synthetic samples look overfit while others remain underfit but still exhibit privacy leakage (see Appendix.FIG.14A. for CDFs of 1-NN distances). PRIVET and In-Authenticity provide binary leak/no-leak predictions at the sample level, allowing direct comparison with ground truth. PRIVET reliably detects privacy leakage, as half of the curves closely follows the ground truth (Fig. 3). In contrast, the In-Authenticity metric shows a mild increase with the fraction of copied samples: it begins with false positives and quickly transitions to underestimating the number of true leaks. Finally, both the generalization gap of FLD and the $C_T$ score trend in the expected direction; however, FLD starts in an overfitting regime, while $C_T$ only signals overfitting when a substantial amount of leakage is present.

Replacing DINOv2 with wavelet packet coefficients Veeramacheneni et al. (2025) further highlight the impact of the embedding on downstream evaluation metrics (see Appendix. Fig. 13). While PRIVET is modality-agnostic by construction, its performance naturally reflects the quality of the chosen embedding. In practice, we recommend inspecting the 1-NN CDFs, which provide immediate feedback on the regime (underfitting or overfitting) induced by a given embedding. This visual diagnostic also helps ensure that PRIVET operates in a representation where its distance-based assumptions are valid. In practice, some data types benefit from an embedding space while others do not. The choice of representation is left to the practitioner, who should select the space that best captures the relevant structure of the data.

## 4 MEMBERSHIP ATTACKS

Here we consider a scenario where we have a large public dataset and synthetic data generated by a private model, for which we want to identify in the public set which examples have been used to train the model. We performed the attack on two datasets: the pseudo-synthetic genetic data where ground truth is known (c.f. Subsection.3.1) and a real-case scenario consisting of RBM generated genetic data. For the latter, an

RBM  Bereux et al. (2025) was trained on 3500 training samples of 805 highly differentiated SNPs  Colonna et al. (2014) coming from  The 1000 Genomes Project Consortium et al. (2015). We intentionally retained generated samples from different epochs corresponding to underfitting, well-fitting, or overfitting stages, as confirmed by their eCDFs (see Appendix.F). In both cases, PRIVET was used to assign a score ($\Delta\pi_r$ (Eqs. 2) was thresholded with $\tau = -3$.) to each sample in the reference data created by merging train and validation sets (see Fig. 4). The analysis of anomalous NN distances is done, this time, on reference-to-synthetic NN against reference-to-reference NN. We label training samples as positive instances, and validation samples as negative ones.

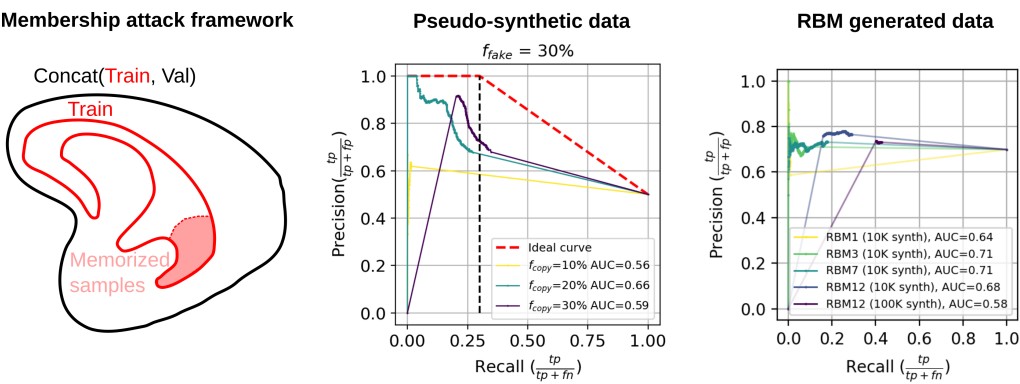

Figure 4: **Membership attack.** The reference data is created by merging train and validation set. The goal is to identify the training samples among the overall database. If the generative model memorizes only a subset of the training data, the true recall reflects the actual proportion of training samples that have been memorized. **Pseudo-synthetic data.** Precision-Recall curve of the membership attack for pseudo-synthetic genetic data. Train, validation and synthetic set have 1668 samples each. 30% of synthetics are leaked, with 10%, 20% and 30% of SNPs copied from train. Vertical black dashed bar corresponds to true recall equal to 30%. Red dashed curve corresponds to an ideal classifier, making random predictions once all memorized samples have been identified. **RBM generated data.** Precision-Recall curve for synthetic data generated at different RBM training stages exhibit distinct regimes: RBM1 and RBM3 show underfitting, RBM7 aligns with the train–train distribution, and RBM12 indicates overfitting. For RBM12, we generated either 10K synthetic or 100K to improve the lower bound of the observed recall.

For a reasonable amount of copied SNPs, PRIVET's recall curves level off around the true memorization fraction, demonstrating that it recovers most leaked samples without significantly overestimating the ground truth leak rate, while maintaining a precision of $\sim 70\%$. When applied to RBM-generated data, where the exact number of memorized training samples is unknown, PRIVET's membership attack exhibits a consistent increase in recall as the model transitions from underfitting to overfitting regime. Additionally, generating more synthetic samples improves recall with minimal impact on precision, which remains around 75%. A comparison with the other metrics (Table 1) within this membership inference framework was not feasible. Among them, only FLD provides a real-valued memorization score at the sample level, though it was not originally evaluated for scoring training samples instead of synthetic ones.

## 5 DIFFERENTIAL PRIVACY

We further assess the sensitivity of PRIVET to controlled levels of privacy protection by evaluating it on synthetic data generated by a differentially private generator. Differential privacy (DP) provides principled guarantees on the upstream generating process and enables regulation of the amount of training information instilled in the model updates.

We first trained a DDPM Ho et al. (2020) on a subset of 1,000 CIFAR10 images for 50 epochs, and generated 1,000 samples, yielding a scenario with privacy leakage in the underfitting regime. This non-private baseline serves as a reference leakage setting, on which DP is subsequently applied to SGD using Opacus Yousefpour et al. (2021). Precisely, we applied DP-SGD with several noise multipliers $\sigma$, producing privacy levels ranging from strong ($\epsilon \approx 1$) to very weak ($\epsilon >> 1$). Details are provided in Appendix K.

As shown in Fig. 5, PRIVET ($n_{\text{pleaks}}(r)$, Eqs. 9) exhibits a clear monotonic trend: stronger DP noise (larger $\sigma$, smaller $\epsilon$) reduces the number of detected privacy leaks and progressively narrows the gap between synthetic-to-train and synthetic-to-test 1-NN distances (Appendix.Fig. 16), while weaker DP rapidly increases NPL toward the non-DP baseline. These results demonstrate that PRIVET detects actual privacy risk and not just overfitting and that PRIVET's privacy score strongly correlates with a controlled privacy parameter $\epsilon$.

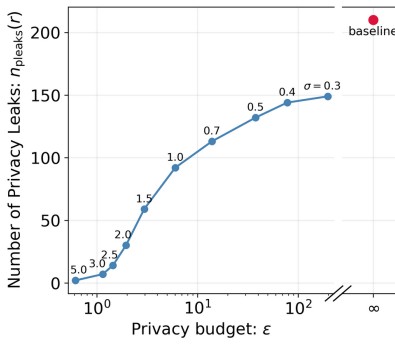

Figure 5: **DP-DDPM on CIFAR10 (1K subset).** Number of detected privacy leaks as a function of $\epsilon$. A DDPM was trained for 50 epochs under DP-SGD (Opacus) with noise multipliers $\sigma$ and $\delta = 10^{-3}$, each generating 1,000 synthetic samples. The test set has 1,000 samples and all seeds are fixed so that Opacus noise is the only randomness. The non-private baseline (red) is trained without DP-SGD and shows leakage in the underfitting regime. More details in Appendix. K

## 6 LIMITATIONS

PRIVET, similar to the other privacy metrics, relies on having a good representation of the data such that distances between data points are meaningful. This is relatively straightforward for genetic data since variation between samples is defined by point mutations on the sequences, allowing simple Hamming distance to be utilizable directly. For image data, however, since different embeddings capture different contextual characteristics from the pixel space, performance of the privacy metrics can change drastically based on the preferred representation (Fig. 3). This said, we show in App. C.1 an asymptotic EVT equivalence property for distances derived from norms. See App. C.2 for further discussion about distance choice impact. Finally, when private information contributes only marginally to the overall distance, the choice of distance metric becomes critical. In such cases, one possible strategy to mitigate this limitation is to adopt a metric that assigns greater weight to sensitive features of the data.

## 7 CONCLUSION

In this work, we presented a novel framework based on extreme value theory, PRIVET [1], for detecting memorization, overfitting and privacy leakage in underfitting regime, in synthetic datasets. Our method can provide both sample-level and dataset-level quantitative assessment of privacy leaks from the training dataset, with the further possibility of assigning a boolean "leak-or-not" flag for each synthetic sample. We tested PRIVET for genetic and image data domains under controlled settings, where the amount of privacy leak is known, and realistic settings with membership inference attacks on samples created by generative models. We compared our method to existing ones and reported substantially better performance under various settings. The asymptotic behavior of nearest-neighbor distances follows the expected Extreme Value regime not only for images and genetic sequences but also for heterogeneous tabular data, as shown in App.Fig. 6. To the best of our knowledge, our method is the first to offer a comprehensive assessment of both individual samples and entire datasets, while remaining theoretically grounded, scalable, and robust across low-sample regimes and diverse domains.

---

[1]https://anonymous.4open.science/r/PRIVET-DF55/

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

## A    NEAREST NEIGHBOR DISTANCES AND EVD

Let us justify more precisely here the fit which is performed on NN distributions. Taking the notations of Section.2.1 we want to characterize the distribution of the random variable $\delta(\boldsymbol{x}, N)$ where $\boldsymbol{x} \sim \rho$. Let us call

$$F_{\boldsymbol{x}}(u) \stackrel{\text{def}}{=} P_{\boldsymbol{y} \sim \rho}\big[d(\boldsymbol{x}, \boldsymbol{y}) < u\big],$$

$$
\begin{aligned}
F_{\boldsymbol{x}}(u, N) &\stackrel{\text{def}}{=} P_{\mathcal{T}_N \sim \rho^N}\big[\delta(\boldsymbol{x}, N) < u\big] \\
&= P_{\mathcal{T}_N \sim \rho^N}\big[\min_{\boldsymbol{y}' \in \mathcal{T}_N} d(\boldsymbol{x}, \boldsymbol{y}') < u\big] \\
&= 1 - P_{\mathcal{T}_N \sim \rho^N}\big[\min_{\boldsymbol{y}' \in \mathcal{T}_N} d(\boldsymbol{x}, \boldsymbol{y}') \geqslant u\big] \\
&= 1 - P_{\boldsymbol{y}_1 \sim \rho, \dots, \boldsymbol{y}_N \sim \rho}\big[\forall \boldsymbol{y}' \in \{\boldsymbol{y}_i\}_{i=1}^N, \ d(\boldsymbol{x}, \boldsymbol{y}') \geqslant u\big] \\
&= 1 - \big(P_{\boldsymbol{y} \sim \rho}\big[d(\boldsymbol{x}, \boldsymbol{y}) \geqslant u\big]\big)^N \\
&= 1 - \big(1 - F_{\boldsymbol{x}}(u)\big)^N .
\end{aligned}
$$

We have

$$F(u, N) = \mathsf{E}_{\boldsymbol{x} \sim \rho}\big[F_{\boldsymbol{x}}(u, N)\big] = 1 - \mathsf{E}_{\boldsymbol{x} \sim \rho}\Big[\big(1 - F_{\boldsymbol{x}}(u)\big)^N\Big] .$$

Since by definition

$$F(u) = \mathsf{E}_{\boldsymbol{x} \sim \rho}\big[F_{\boldsymbol{x}}(u)\big]$$

by convexity of the function $x \to (1-x)^N$ we get

$$F(u, N) \leqslant 1 - \big[1 - F(u)\big]^N$$

with equality when the function $u \mapsto F_{\boldsymbol{x}}(u)$ does not depend on $\boldsymbol{x}$. Hence **in the case of a uniform distribution over a manifold, this bound is an equality**. Indeed in the limit when the number $N$ of samples tends to infinity, the distribution of distances to nearest neighbors is the same at all locations $\boldsymbol{x}$: every point $\boldsymbol{x}$ has the same density and a neighborhood of same dimensionality, which is locally identical in the infinitesimal limit (tangent space).

In the case of more complex distributions, let us call

$$\bar{G}(v) \stackrel{\text{def}}{=} \lim_{N \to \infty} \big[1 - F(u)\big]^N$$

with $v$ the appropriate rescaled variable, depending on the universality class of the EVD. Indeed, depending on the behavior of the lower tail of the distribution $F$, the corresponding extreme value statistics either converges to a Weibull distribution:

$$G(v) = \exp(-Av^{\alpha}), \tag{6}$$

with $a_N = N^{-1/d}$ and $b_N = 0$, when $F(u)$ has typically a power law tail with exponent $\alpha$; either to a Gumbel distribution if for instance $F(u)$ has a Gaussian tail again at small $u$:

$$G(v) = \exp\big(-e^v\big), \tag{7}$$

after centering and normalizing the relative distance variable $u \to \tilde{u} = (u - \bar{u})/\sigma$ w.r.t. to mean $\bar{u}$ and std $\sigma$, with a scaling $v = (\tilde{u} + b_N)/a_N$ now given by $b_N = \sqrt{2 \log(N)} + \mathcal{O}\big(\log \log(N)\big) = 1/a_N$ see Haan & Ferreira (2006).

So as we see $F(u, N)$ is asymptotically a mixture of EVD, indexed by $\boldsymbol{x}$, which is dominated by the EVD defined by the averaged pairwise distance distribution. To see how this mixture behaves, first we make the

hypothesis of a data manifold of dimension $d$ so that the average over $\boldsymbol{x}$ and $\boldsymbol{y}$ to get $F(u)$ is to be understood as

$$F(u) = \int d\boldsymbol{x}^d d\boldsymbol{y}^d \rho(\boldsymbol{x})\rho(\boldsymbol{y})\mathbf{1}_{\{d(\boldsymbol{x},\boldsymbol{y})\leq u\}},$$

where $\rho$ is to be considered as the probability density of the data on this manifold, and $d(\boldsymbol{x},\boldsymbol{y})$ a distance induced by a metric on the manifold. The metric is chosen in such a way that the expected number of samples inside a sphere of infinitesimal radius $\epsilon$ centered on $\boldsymbol{x}$ scales like $\rho(\boldsymbol{x})\epsilon^d$, meaning in practice that the length on all directions are measured in comparable units. With these hypothesis we basically have that when sampling $\boldsymbol{y} \sim \rho$ the radial probability density w.r.t. some point $\boldsymbol{x}$ is locally given by the distribution

$$f_{\boldsymbol{x}}(u) = P_{\boldsymbol{y}\sim\rho}[d(\boldsymbol{x},\boldsymbol{y}) \in du] \propto \rho(\boldsymbol{x})du u^{d-1}e^{-V_{\boldsymbol{x}}(u)}$$

where $V_{\boldsymbol{x}}$ is in principle dependent on $\boldsymbol{x}$, but we assume it to be a smooth potential with $V(u) \to \infty$ at large $u$ and to continuous and differentiable when $u$ is close to zero.

**Power law tail:**   For $d = \mathcal{O}(1)$, the corresponding probability density behave like $F_{\boldsymbol{x}}(u) \sim \rho(\boldsymbol{x})u^d$ at small $u$, hence when $N$ get sufficiently large, then the NN neighbor distance to $\boldsymbol{x}$ when sampling $N$ samples $\boldsymbol{y} \sim \rho$ will scale like $u_N \sim [N\rho(\boldsymbol{x})]^{-1/d}$, leading us to a Weibull distribution with exponent $\alpha = d$ independent of $\boldsymbol{x}$ and $a_N \sim [N\rho(\boldsymbol{x})]^{-1/d}$. So, with these hypothesis we expect the coefficient $\hat{A}$ of the Weibull fit (1) to depend on $\rho(\boldsymbol{x})$ but not the exponent.

**Gaussian tail:**   Instead when $d \gg 1$, $f_{\boldsymbol{x}}$ becomes Gaussian around the point $\tilde{u}$ satisfying $V'_{\boldsymbol{x}}(\tilde{u}_{\boldsymbol{x}}) - (d-1)/\tilde{u}_{\boldsymbol{x}} = 0$. So we then get a Gaussian tail, leading us to a Gumbel distribution, but in principle both $\tilde{u}_{\boldsymbol{x}}$ and the standard deviation $\sigma_{\boldsymbol{x}}$ should depend on $\boldsymbol{x}$. If this dependency can be neglected, then the coefficient $\hat{B}$ in the Gumbel fit (1) is independent of $\boldsymbol{x}$. We will assume that in the following. By contrast, the other coefficient $\hat{A}$ should depend in any case on $\rho(\boldsymbol{x})$. So considering now $F(u, N)$ when $N$ is large under the hypothesis of a data manifold we have

$$\lim_{N\to\infty} F(u, N) = \int d^d\boldsymbol{x}\rho(\boldsymbol{x})[1 - G_{\boldsymbol{x}}(v)]$$

If we look at the tail, i.e. $v \to 0$ for Weibull and $v \to -\infty$ for Gumbel, we have

$$\lim_{N\to\infty} F(u, N) \approx \begin{cases} \bar{A}v^\alpha, & \text{(Weibull case)} \\ \bar{A}e^{Bv}, & \text{(Gumbel case)} \end{cases}$$

with

$$\bar{A} = \mathsf{E}_{\boldsymbol{x}\sim\rho}\left[A_{\boldsymbol{x}}\right]$$

In addition, we have the fact that in both cases the coefficient $A_{\boldsymbol{x}}$ inherits his dependency on $\boldsymbol{x}$ from $F_{\boldsymbol{x}}(u)$ which is of the form when $u$ is close to zero

$$F_{\boldsymbol{x}}(u) \approx \rho(\boldsymbol{x}) \begin{cases} u^\alpha, & \text{(power law tail)} \\ \displaystyle\int_{-\infty}^u dr e^{-\frac{(r-\tilde{u})^2}{\sigma^2}}, & \text{(Gaussian tail)} \end{cases}$$

As a result, the average which is done beforehand in $\bar{G}$ leads to the same constant fit $\bar{A}$ which somewhat justifies (given the necessary hypothesis) the fit we are doing.

## B  VARIOUS SCORES BASED ON ORDER $r$TH STATISTICS

$\pi_r^{\text{ref}} = P_{N,M}(u, r)$ is the key statistic that we use to detect overfitting and privacy leaks. In Section 2.3 we propose one straightforward individual privacy score

$$\Delta\pi_r = \log \frac{\pi_r^{\text{train}}}{\pi_r^{\text{test}}}$$

which yields good results, but other possible definitions exist. The rational behind this log ratio is to condition the order $r$th statistics for the train on the $r$th statistics for the test

$$\frac{\pi_r^{\text{train}}}{\pi_r^{\text{test}}} = P(\delta_r(N, M) < u_r^{\text{train}} | \delta_r(N, M) < u_r^{\text{test}}),$$

where $\delta_r(N, M)$ is the random variables corresponding to the ranked $r$th NN distance among $M$ synthetic data from a reference set of size $N$, following $G(u)$, while $u_r^{\text{train,test}}$ are the value of these distances actually observed respectively w.r.t. the train and to the test set. This obviously make sense when $u_r^{\text{train}} < u_r^{\text{test}}$ which is most generally observed.

A very small value of $\pi_r^{\text{ref}}$ is telling us that the $r$th NN distance $u_r$ is too small, or stated differently there are too many samples with NN distance below $u_r$. In contrary a very small value of $1 - \pi_r^{\text{ref}}$ that there are too few samples with NN distance below $u_r$. $\Delta\pi_r$ is therefore meaningful mostly in the over-parameterized regime, because in the under-parameterized regime both probabilities $\pi_r^{\text{train,test}}$ rapidly saturate to one when the tail of the NN distribution of $d_{STr}^*$ and $d_{STe}^*$ are well below the expected one obtained from $d_{TrTr}^*$. A simple extension then consists to consider also

$$\overline{\Delta\pi_r} = \log \frac{1 - \pi_r^{\text{test}}}{1 - \pi_r^{\text{train}}}$$

which will as well indicate privacy leaks in the underfitting regime when very negative. In addition we expect the excess or deficit of sample below $u_r$ to have multiplicative effect on the probability $\pi_r^{\text{ref}}$, so taking the $\log$ should scale linearly, hence qualitatively we expect that both $\Delta\pi_r$ and $\overline{\Delta\pi_r}$ should scale linearly with the number of privacy leakage up to rank $r$, respectively in the over- and under-fitting regime.

A more refined quantitative estimation of the number of overfiting samples or privacy leaks can be obtained as follows. For a given distance $u$ we can obtain this by comparing the actual rank observed for this distance with the rank $r(u)$ corresponding to

$$P_{N,M}(u, r(u)) = \frac{1}{2},$$

the difference between the two giving the excess (or deficit if negative) of samples having NN distance below $u$. For large $M$ this rank is directly read off from $\hat{G}$:

$$r(u) \approx M\hat{G}(u) \qquad (r(u) \gg 1)$$

The reason is that the sum of binomial terms appearing in the definition of $P_{N,M}(u, r)$:

$$P_{N,M}(u, r) \stackrel{\text{def}}{=} 1 - \sum_{q=0}^{r-1} \frac{M!}{q!(M-q)!} G(u)^q \left[1 - G(u)\right]^{M-q}$$

when non-vanishing, becomes concentrated on a few terms when $M \gg 1$, the maximum being obtained from the asymptotics

$$\frac{M!}{q!(M-q)!} G(u)^q \left[1 - G(u)\right]^{M-q} \approx \exp\left(-M\left[\nu_q \log \frac{\nu_q}{G(u)} + (1 - \nu_q) \log \frac{1 - \nu_q}{1 - G(u)}\right]\right)$$

with $\nu_q = q/M$. Since these terms are symmetric under the exchange $q \leftrightarrow M - q$ the maximum term obtained for $\nu_r \approx G(u)$ corresponds to the rank $r$ s.t. $P_{N,M}(u, r) = \frac{1}{2}$. As a result $r_u - r(u)$ represents the excess of samples having NN distance below $u$ if $r$ is the rank of the sample having NN distance u. This leads to define the following quantities:

$$n_{\text{overfit}}(r) \stackrel{\text{def}}{=} r - r[u_r^{\text{train}}] \tag{8}$$

$$n_{\text{pleaks}}(r) \stackrel{\text{def}}{=} r[u_r^{\text{test}}] - r[u_r^{\text{train}}] \tag{9}$$

where $u_r^{\text{train}}$ and $u_r^{\text{test}}$ represents the $r$th NN distance respectively to the train and to the test dataset. $n_{\text{overfit}}(r)$ provides us directly with an estimation, among the first $r$ synthetic samples of lowest NN distance to the train, of the expected number of overfitting samples; $n_{\text{pleaks}}(r)$, by making use of the NN distance corresponding to the same rank $r$, w.r.t. the test set, actually measures the excess of small NN distances to the train, compared to the test, among the same first samples $r$. From these quantities we can obtain individual probability scores as follows. If we denote by $n_{\text{excess}}(r) \stackrel{\text{def}}{=} r - r[u_r^{\text{ref}}]$ the expected excess of samples up to rank $r$ whatever reference set is considered (either train of test), then by correcting of the rank $r$ the excess $n_{\text{excess}}(r - 1)$ we get an estimation of the individual influence of the sample $r$ to the order $r$th statistics which we turn into the probability score

$$p_r^{\text{ref}} = \pi_{r - n_{\text{excess}}(r-1)}^{\text{ref}}$$

From this we define the individual overfitting score as $p_r^{\text{train}}$ and the individual privacy leak score

$$\Delta p_r \stackrel{\text{def}}{=} \log \frac{p_r^{\text{train}}}{p_{r'}^{\text{test}}} \tag{10}$$

where now we presumably want to rather use the rank $r'$ of $\boldsymbol{x}_r$ in the NN distribution w.r.t. the test set, rather than $r$ itself since the conditional probability interpretation does not hold in this case.

## C  ABOUT CHOOSING DISTANCES

### C.1  ASYMPTOTIC EQUIVALENCE OF NORM-DERIVED DISTANCES

While the choice of a meaningful distance is *a priori* crucial, we show nonetheless that in the aymptotic regime with infinitely many samples, the slope of the fit computed by our method based on Extreme Value Theory does not depend on the choice of the distance, provided it derives from a norm.

**Proof**

In finite dimension $p$ indeed, all norms are equivalent, that is, for any two distances $d$ and $d'$ derived from norms on $\mathcal{X} = \mathbb{R}^p$,

$$\exists \alpha, \beta > 0, s.t. \forall \boldsymbol{x}_1, \boldsymbol{x}_2 \in \mathcal{X}, \quad \alpha \, d(\boldsymbol{x}_1, \boldsymbol{x}_2) \leqslant d'(\boldsymbol{x}_1, \boldsymbol{x}_2) \leqslant \beta \, d(\boldsymbol{x}_1, \boldsymbol{x}_2)$$

This has consequences on nearest-neighbor distances: given a dataset $\mathcal{D} = \{\boldsymbol{x}_i\}_{1 \leqslant i \leqslant n}$ of points in $\mathcal{X}$, the distances of any given point $\boldsymbol{x} \in \mathcal{X}$ to the closest neighbor in the set $\mathcal{D}$ are similar. Indeed, noting $\delta(\boldsymbol{x}) = \min_{i \in \mathcal{D}} d(\boldsymbol{x}, \boldsymbol{x}_i)$ and $\delta'(\boldsymbol{x}) = \min_{i \in \mathcal{D}} d'(\boldsymbol{x}, \boldsymbol{x}_i)$, and $\boldsymbol{x}^*, \boldsymbol{x}^{*'}$ the associated closest points for $d$ and $d'$ respectively, we have:

$$\alpha \, \delta(\boldsymbol{x}) = \alpha \, d(\boldsymbol{x}, \boldsymbol{x}^*) \leqslant \alpha \, d(\boldsymbol{x}, \boldsymbol{x}^{*'}) \leqslant d'(\boldsymbol{x}, \boldsymbol{x}^{*'}) = \delta'(\boldsymbol{x}) \leqslant d'(\boldsymbol{x}, \boldsymbol{x}^*) \leqslant \beta \, d(\boldsymbol{x}, \boldsymbol{x}^*) = \beta \, \delta(\boldsymbol{x})$$

hence $\alpha \, \delta(\boldsymbol{x}) \leqslant \delta'(\boldsymbol{x}) \leqslant \beta \, \delta(\boldsymbol{x})$, and in particular:

$$\log \alpha + \log \delta(\boldsymbol{x}) \leqslant \log \delta'(\boldsymbol{x}) \leqslant \log \beta + \log \delta(\boldsymbol{x}) \, .$$

The histograms of nearest neighbor distances are thus quite similar, as histogram mass can be at most shifted by bounded values, namely $\log \alpha$ and $\log \beta$.

Now, if the range of distances (in log scale) on which the linear fit is performed (see Figure 1) is arbitrary large, the consequences of such bounded shifts on the fitted slope become arbitrarily negligible as well.

It turns out that when the number $M$ of samples increases, the nearest-neighbor distance histogram can be refined (since more points allow for more precise density estimation), and the region where the linear fit is performed gets extended towards smaller values (in log scale). More precisely, this region is proportional to $\left[\frac{1}{M}, q\right]$ where $q$ is a fixed quantile and $\frac{1}{M}$ is the smallest quantum of probability available (corresponding to the weight of one sample in the dataset); in log scale this yields $[-\log M, \log q]$, a range that tends to infinity when the number of samples does as well. Q.E.D.

## C.2 DISCUSSION ABOUT DISTANCES

While all norms satisfy the property above, note that this is not the case of semi-norms. Indeed semi-norms yield 0 distance in certain directions, which implies $\alpha = 0$ and/or $\beta = +\infty$, breaking the equivalence and the asymptotic property. Note also that not all distances can be derived from norms.

The choice of a meaningful distance is crucial, as the privacy leaks are perceived under that lens. This said, as shown by the equivalence aymptotic property under certain conditions, EVT can detect leaks beyond the original intention: for instance in our application case with DNA and Hamming distance between SNPs, one could expect to detect only generated samples that are too close to one training sample (i.e. most SNPs are identical). In the meanwhile, an important concept in population genetics is the one of recombination (copying a part from an individual and then another part from another one). This yields a notion of similarity between individuals that does not seem related to the Hamming distance at first glance, as a recombination between 2 samples is far from each of the samples for that distance, in terms of number of SNPs that differ. However, a recombination is still visible in term of Hamming distance, as the distance to its original individuals is halved, w.r.t. typical distances (as half of the SNPs are identical). Statistically, in high dimensions, this is not probable, and will consequently be detected by the EVT approach.

## D    ASSESSING THE QUALITY OF FITS TO THE NN DISTRIBUTIONS

### D.1    GOODNESS-OF-FIT

We perform a meta-analysis across datasets and modalities to empirically assess the heavy-tail assumption. More specifically, we assess whether the empirical distribution of nearest-neighbor distances in each dataset is consistent with extreme value distributions. The procedure for **FIG.6.A** is as follows:

1. **Nearest-neighbor distances.** For each dataset, we compute 1-NN distances between $N$ train samples (leading to train–train distances) using the appropriate feature representation (images: dinov2 embeddings, SNPs: remains binary, tabular features: categorical variables are one-hot encoded and numerical ones are standardized). Distances computed with the appropriate distance (euclidean for images and tabular data, hamming for SNPs) are sorted into increasing order, yielding

$$\delta_{(1)} \leq \delta_{(2)} \leq \cdots \leq \delta_{(N)}.$$

2. **Tail windows.** A lower cutoff fraction $a$ removes very small distances (unreliable statistical fluctuations of the tail, typically set to $1\%$ by default), and an upper quantile $q$ specifies the end of the tail window on which the EVT fit will be applied. The support of the resulting double-truncated distribution is

$$[a, b_q] = \left[ \delta_{(\lfloor a \cdot N \rfloor)}, \ \delta_{(\lfloor qN \rfloor)} \right]$$

containing $m$ tail distances. We here made use of the abuse of notation that $a$ and $b_q$ represents the cutoff fractions but also the $\lfloor a \cdot N \rfloor$-, resp. $\lfloor q \cdot N \rfloor$-, order variable.

3. **Truncated maximum likelihood fits.** On this restricted interval, we fit Weibull and Gumbel models via truncated maximum likelihood estimation (MLE). Truncation means the likelihood is conditioned on all data lying in $[a, b_q]$, so the fitted distribution is normalized to that window. Between Weibull and Gumbel, the family with the smaller negative log-likelihood (NLL) is selected.

4. **Probability Integral Transform (PIT).** Using the fitted CDF $\hat{F}$ on that restricted interval, we compute the PIT H. (1932); Rosenblatt (1952) for each sample falling in that interval

$$v_i = \frac{F(\delta_i) - F(a)}{F(b_q) - F(a)} \quad \in [0, 1].$$

Under a correct model, the $v_i$ should be approximately i.i.d. Uniform$[0, 1]$.

5. **P-P plots.** For each $q$ in a range between $0.1$ and $0.3$, we compute the empirical CDF for the PIT variables

$$\hat{G}_q(u) = \frac{1}{m} \sum_{i=1}^{m} \mathbf{1}\{v_i \leq u\}, \quad u \in [0, 1],$$

and plot $\hat{G}_q(u)$ against the uniform line $u$. The blue ribbon in **FIG. 6.A.** shows the variability of tail fits across the range of $q$-quantiles.

6. **Parametric bootstrap KS band.** To quantify deviation from uniform distribution, we compute the Kolmogorov–Smirnov (KS) statistic

$$D_q^{\text{obs}} = \sup_{u \in [0,1]} \left| \hat{G}_q(u) - u \right|$$

for a fixed reference quantile taken to default value $q = 20\%$ (orange curve). Because parameters are estimated from the same data, the distribution of $v_i$'s are no long independent uniforms (David & Johnson, 1948). We thus use a parametric bootstrap to construct a corrected model:

- we simulate $m$ synthetic samples from the fitted truncated model via the inverse transform sampling or inverse PIT
- we refit the EVD model on each synthetic windows via MLE;
- we recompute PITs and their KS statistic for each bootstrap $D_q^{\text{bootstrap}_j}$.

The 95-th percentile of the KS statistics over these bootstrap replicates (200 bootstraps) provides a calibrated critical value that defines the width of the KS band for the fixed partition $[a, b_q]$ (orange band). The observed statistic $D_q^{\text{obs}}$ corresponding to orange curve is reported, along with its Monte-Carlo $p$-value, defined as the fraction of bootstrap replicates exceeding $D_{\text{obs}}$. The parametric bootstrap band is thus specifically calibrated for a single eCDF curve corresponding to $q = 20\%$, but doesn't generalize for the blue ribbon.

The result is a diagnostic P-P plot: the blue ribbon shows the variability of fits across quantiles (a hyperparameter of the fit varying inside a fixed interval, from which we generate the eCDF of the PIT $q \mapsto \hat{G}_q$), the orange curve shows a representative fit for the default quantile, and the shaded orange band around it shows the bootstrap calibrated acceptance region for the corresponding KS statistic testing the following null hypothesis: the data in $[a, b_{q=20\%}]$ come from the EVD fit. The width of the KS band scales as $\frac{1}{\sqrt{m}}$.

We first observe that the orange curve $\hat{G}_{q=20\%}(u)$ almost always closely follows the diagonal $y = x$, with the exception of the UCI Adult dataset Becker & Kohavi (1996), which exhibits small but visible deviations. The parametric bootstrap KS test rejects the null hypothesis of uniformity for CelebA Liu et al. (2015), UCI Adult, and UCI Bank Moro et al. (2014). However, this rejection is due to the inflated sensitivity of the KS statistic at large sample sizes. Importantly, the narrow blue ribbon in **Fig. 6.A** shows that the diagnostic is robust to the choice of the tail hyperparameter $q$. Overall, **Fig. 6.A** supports the conclusion that the EVT tail assumption is a general one and holds in all scenario tested.

## D.2 CONSISTENCY ACROSS EQUIVALENT SPLITS

To assess whether the fitted tail is stable across equivalent partitions of the data, we perform repeated random splits of the training dataset into two halves of equal part (**FIG.6.B**). On each split, we:

1. Fit the best candidate distribution (Weibull or Gumbel) on one half ("train1") restricted to the default tail window $[a, b_q]$ for a fixed quantile $q = 20\%$ and $a = 1\%$. The fitted CDF $\hat{F}$ is then frozen.

2. Apply this frozen $\hat{F}$ to the other half ("train2"), computing PITs

$$v_i = \frac{\hat{F}(\delta_i) - \hat{F}(a)}{\hat{F}(b) - \hat{F}(a)}, \quad \delta_i \in [a, b_q].$$

   where $[a, b_q]$ is here a subset in "train2".

3. Construct the eCDF $\hat{G}_q(u)$ of these $\{v_i\}_{i=1\cdots m}$.

This procedure is repeated across many random seeds, yielding a family of eCDF curves (blue ribbon), with the median eCDF overlaid in dark blue. Since each curve arises from a different "train1"-"train2" split of the same underlying dataset, a narrow ribbon centered on the diagonal indicates that the fitted tail is stable across subsets. In contrast, wide ribbons or systematic deviations above/below the diagonal would reveal that the estimated model is sensitive to the particular split, and thus not reproducible, i.e. the model depends strongly on which partition is used for fitting. Empirically, we observe wider ribbons for CIFAR10 and the 65K SNP dataset, although the median eCDF remains close to the diagonal. For the SNP case, this mild instability is plausibly explained by kinship relations between individuals, which can affect NN tail distances. By contrast, the ribbon remains narrow for the remaining datasets and modalities. This diagnostic thus complements the

goodness-of-fit test by assessing the reproducibility of the fitted tail law across equivalent partitions within a given dataset.

This indicates that PRIVET extends naturally to mixed data types (categorical + continuous): in our experiments, euclidean distance on preprocessed features (normalization and one-hot encoding) was sufficient, and alternative distances for heterogeneous data types exist in the literature Meeus et al. (2024).

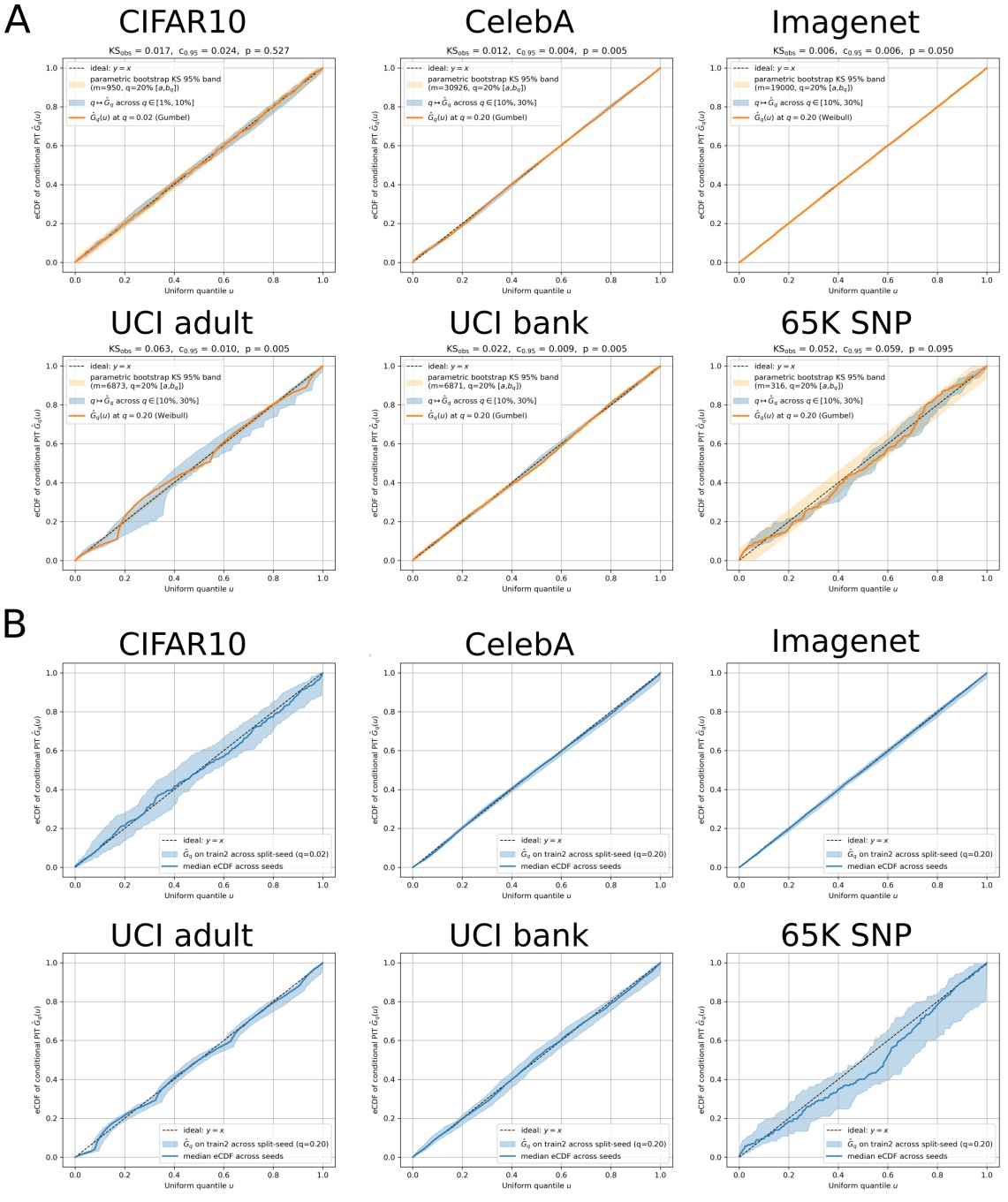

Figure 6: **Meta-analysis across datasets and modalities. A.** Goodness-of-Fit diagnostics. **B.** Consistency of fit between equivalent partitions.

# E CONTROLLED EXPERIMENTS ON SYNTHETIC DATA

## E.1 LOW SAMPLE SIZE

In addition to the experiments described in the main text, we illustrate the robustness of PRIVET in a low-sample-size setting. These data are real genetic data corresponding to binary sequences of size 33602 retrieved from The 1000 Genomes Project Consortium et al. (2015). Distance between samples is simply the Hamming distance. We partition these 198 samples into three random subsets each of equal size 66 and arbitrarily assign these to train, test and synthetic datasets, the latter corresponding to the output from a fictional generative model. The x and y-axis corresponding respectively to $f_{\text{fake}}$ and $f_{\text{copy}}$ are used to construct pseudo-synthetic data as presented in Main. Section. 3.1.

Fig. 7 compares the behavior of the different global scores when both $f_{\text{fake}}$ and $f_{\text{copy}}$ are varied. Global results are not shown for FLD Jiralerspong et al. (2023), PQMass Lemos et al. (2025a) and $C_T$ Meehan et al. (2020) due to execution failures caused by small sample size that cannot be handled by these metrics. We observe limited sensitivity and noisy behavior for Privacy loss score of $\mathcal{AATS}$ in contrast to the scenario where sample size is larger (c.f. Main. Fig. 2). PRIVET recovering the number of privacy leaks (NPL) exhibits strong sensitivity to both axes. A distinct boundary separates regions of no detection from detection, beginning at $18\%$ copied SNPs (from the closest neighbors) and with as few as $0.1\%$ leaked samples (i.e., 1 synthetic samples), and extending to $40\%$ leaked synthetic samples with as little as $4.7\%$ copied SNPs. In-Auth is highly responsive to privacy leakage signals and achieves strong recall, however, this comes at the cost of many false positives, resulting in substantially lower precision than PRIVET in the intermediate and high leakage regimes.

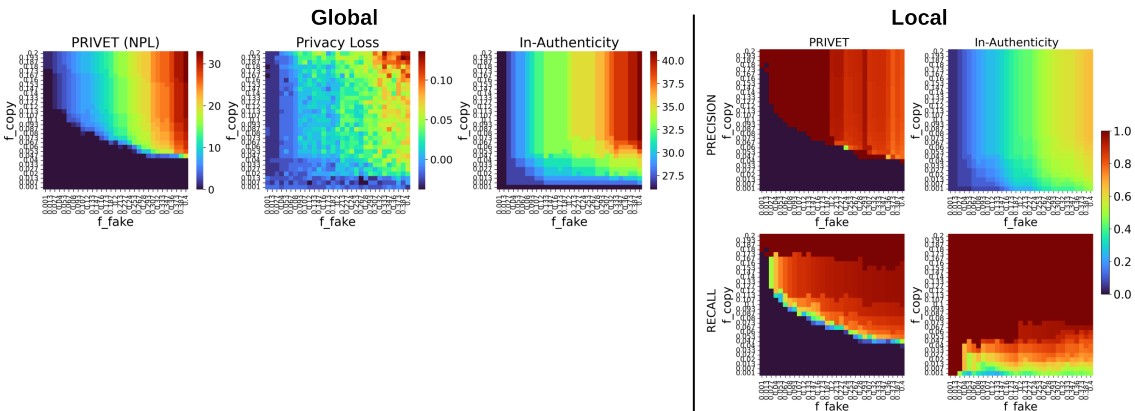

Figure 7: **Global and local privacy scores for 33,602 SNPs and 66 samples. Global:** From left to right—number of privacy leaks (NPL) from PRIVET, generalization gaps of $\mathcal{AATS}$ (i.e. Privacy loss) and In-Authenticity score, plotted over $f_{\text{fake}}$ vs. $f_{\text{copy}}$. **Local:** Precision (top) and recall (bottom) are reported for local privacy metrics.

Dissecting further, we report true/false positives and negatives for the groundtruth, PRIVET (NPL) and In-Authenticity, as these are the only methods providing binary (leak/no-leak) predictions at the sample level (Fig. 8). The first row shows the groundtruth, defined by the values of $f_{\text{fake}}$. In the second row, PRIVET (NPL) exhibits few false positives on second column. In contrast, the false positive map produced by In-Authenticity remains flat along $f_{\text{copy}}$ axis with rates substantially higher than for PRIVET (NPL).

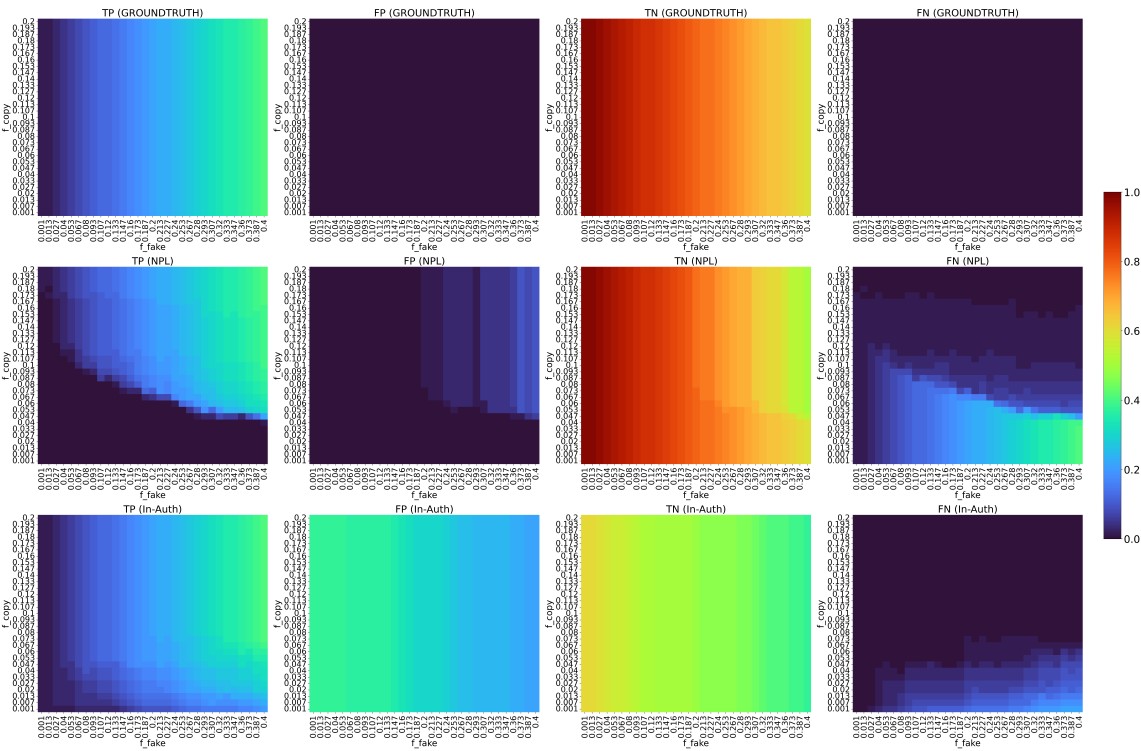

Figure 8: **Classification results for 33,602 SNPs and 66 samples.** The columns represent: True Positives (TP), False Positives (FP), True Negatives (TN), and False Negatives (FN). The rows represent the groundtruth, PRIVET (NPL) and In-Authenticity score. Privacy leaks are reported as a percentage of the synthetic data. $(f_{\mathrm{fake}}, f_c)$ range from $(0.1\%, 0.1\%)$ to $(40\%, 20\%)$.

### E.2   GENETIC DATA

The 1000 Genomes (1KG) Project is a large-scale public resource of human genome sequences designed to represent the global genetic diversity of the human population by sampling individuals from distinct populations across multiple continents The 1000 Genomes Project Consortium et al. (2015). The dataset used in this study comprises 65,535 contiguous SNPs spanning chr1:534247-81813279 (approximately 80 megabase pairs) within the Omni 2.5 genotyping array framework. The data is organized as follows: rows represent phased haplotypes (one of the two chromosome copies inherited from a parent, with known ordering of variants), while columns indicate allele positions, encoded as 0 when the nucleotide matches the reference genome (GRCh37), and 1 when it differs (i.e., a point mutation). This results in a binary matrix of aligned sequences.

### E.3   CLASSIFICATION RESULTS FOR 65,535 SNPS AND 1,668 SAMPLES

In Fig. 9 we display PRIVET (NPL) and In-Auth true positives and false positives maps for the larger sample size genetic dataset (65,535 SNPs and 1,668 samples) Note that the sum of the two maps (TP+FP) yields the

PRIVET (NPL) and In-Auth scores presented in Main.Fig. 2. Once again, we can see the tendency of In-Auth of overestimating privacy leakage.

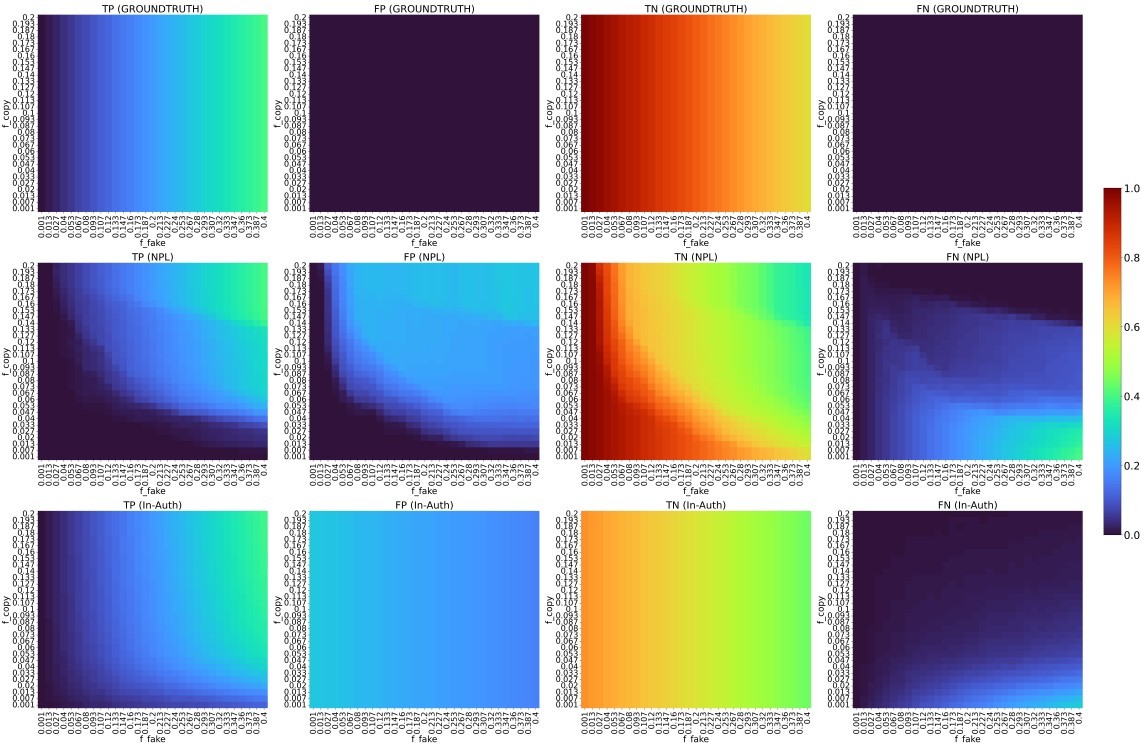

Figure 9: **Privacy maps for 65,535 SNPs and 1668 samples.** The columns represent: True Positives (TP), False Positives (FP), True Negatives (TN), and False Negatives (FN). The rows represent the groundtruth, PRIVET (NPL) and In-Authenticity score. Privacy leaks are reported as a percentage of the synthetic data. $(f_{\text{fake}}, f_c)$ range from $(0.1\%, 0.1\%)$ to $(40\%, 20\%)$.

### E.4 SETTINGS FOR BENCHMARKED METRICS

For all controlled experiments and all metrics, including $C_T$, FLD, Authenticity Score, and the Privacy Loss of $\mathcal{AATS}$, we applied default parameter values. We relied on the official GitHub implementation from Jiralerspong et al. (2023) for all metrics, except PQMass and Privacy Loss, which were implemented separately.

For PQMass, we set `re_tesselation = 1000`, following author's recommendation that higher values yield improved performance.

A normalization preprocessing step is applied internally by FLD. However, when applied to genetic data, this step fails in the presence of fixed sites—SNP positions where all individuals share the same allele (0 or 1)—leading to errors. To address this, we excluded fixed sites from the test data, and then removed the corresponding positions from the train and synthetic sets as well. This resulted in the removal of approximately 200 SNPs per dataset.

For PRIVET, we fitted the NN distances between train-train samples with a Weibull distribution for all datasets. When no mode is observed on the eCDFs, we typically fit on the partition of the data corresponding to the first $1\%$ to $20\%$ of the sorted NN distances. When multi-modal structure is observed, the partition typically ranges from $0.1\%$ to 2 or $5\%$, depending on the specificity of the first mode on which we want to apply the fit.

### E.5 SCORE REFINEMENT

**Decimation**   Although to a substantially lesser extent than In-Auth, PRIVET (NPL) still produces false positives. To address this issue, synthetic samples with privacy scores below the threshold should be sequentially removed, with the scores of the remaining samples recomputed at each step to ensure a proper cleanup of the synthetic data. This additional decimation step adds a cost of $\mathcal{O}(kN)$ where $N$ is the number of synthetic samples and $k$ is the number of detected privacy leaks among them. We compared PRIVET with and without decimation (Fig. 11). The excess of small synthetic to train NN distances due to leakage artificially inflates the remaining eCDF, i.e. it artificially increases the rank of the remaining synthetic samples. This might in turn impact $\Delta\pi_r$ score. By progressively removing the synthetic privacy breaches, we observe a notable improvement in precision, albeit with a decrease in recall. Examining the F1 score, the decimated version of PRIVET shows reduced sensitivity to the copy fraction $f_{\text{copy}}$, while achieving higher scores when both $f_{\text{fake}}$ and $f_{\text{copy}}$ are large. Given the context of privacy leakage detection, we argue that false negatives are more critical than false positives.

**Implicit decimation**   In **FIG.2.Local**, the recall of PRIVET appears to vary with the fraction of leaked samples, which is an undesirable dependency. By contrast, the individual privacy leak score defined in Eq. 10 (Appendix B) corrects for this effect while simultaneously improving upon the raw PRIVET score in **FIG.2.Local**. With this alternative formulation, PRIVET achieves performance comparable to In-Auth (**FIG. 10**), creating smoother detections in the low $(f_{\text{fake}}, f_{\text{copy}})$ region. Nonetheless, turning this continuous score into binary "leak" or "not leak" flag requires careful threshold fine-tuning. In our experiments, we used a threshold of $-0.00115$ corresponding to a pareto front on a Precision-Recall plot.

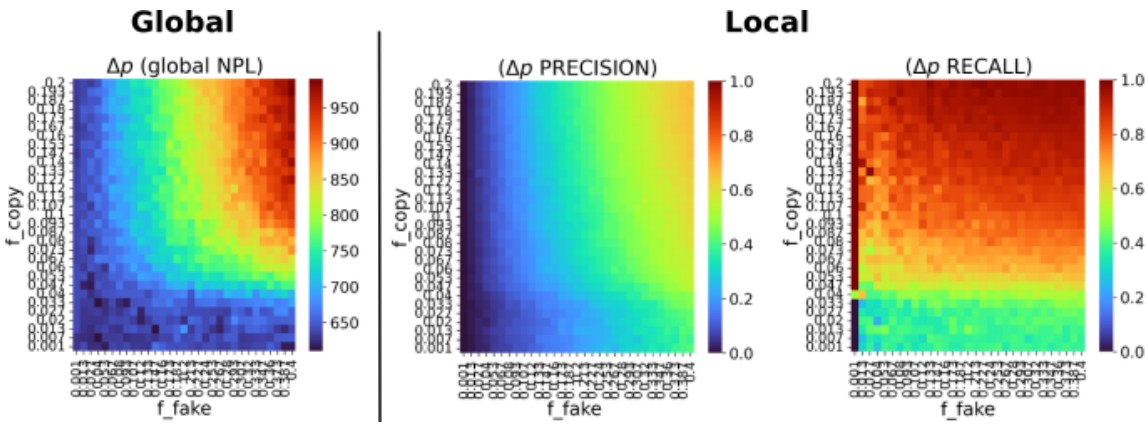

Figure 10: **PRIVET local scores with implicit decimation (Eq. 10).** Controlled experiment on genetic data: 65,535 SNPs and 1,668 samples.

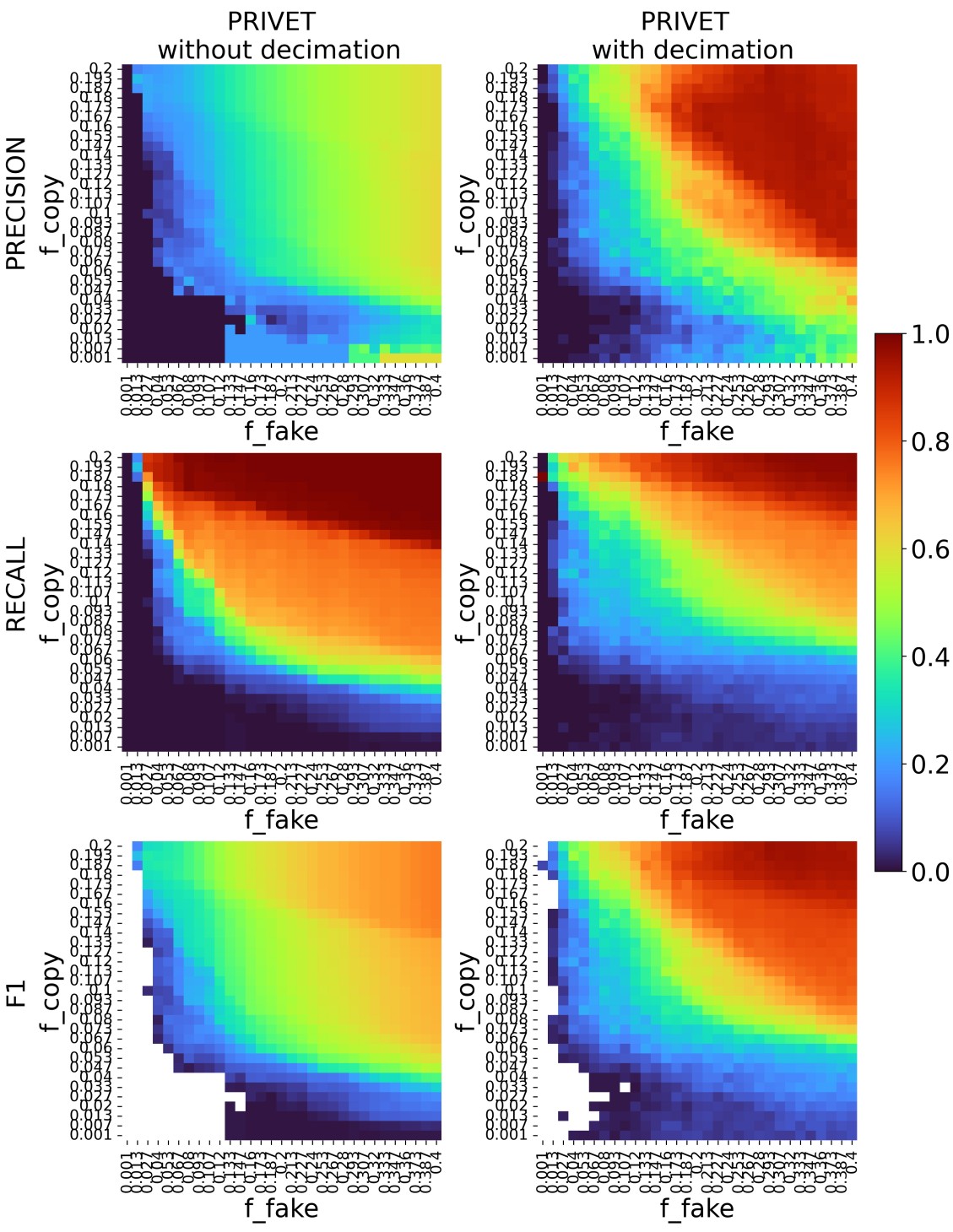

Figure 11: **PRIVET local scores without and with decimation.** Controlled experiment on genetic data: 65,535 SNPs and 1,668 samples. Rows corresponds to precision, recall and F1 score respectively, plotted over $f_{\text{fake}}$ vs. $f_{\text{copy}}$. Columns correspond to PRIVET without decimation (see local privacy maps in Main.Fig. 2 and PRIVET with decimation. White pixels in F1 score correspond to nan value where precision and recall are both equal to 0.

## F  RBM TRAINING

The RBM model for the Human Genome Dataset (HGD) obtained from the 1000Genomes Project was trained using the method proposed in Bereux et al. (2025). The hyperparameters are displayed in table 2

| Dataset | HGD |
|---|---|
| Algorithm | RCM+PTT |
| #RCM hidden nodes | 200 |
| #RBM hidden nodes | 300 |
| #Gibbs steps | 20 |
| learning rate | 0.01 |
| #chains | 2000 |
| batch size | 2000 |

Table 2: Hyperparameters used for the training of the RBM. The number of RCM hidden nodes correspond to the maximum amount before performing decimation.

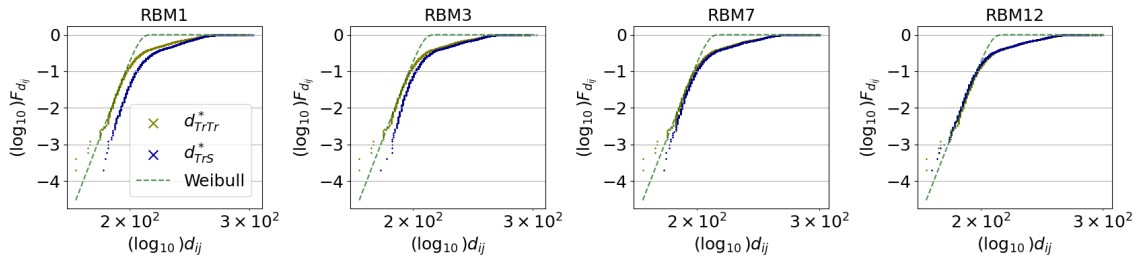

Figure 12: **Evolution of the eCDFs during RBM training.** $d^*_{TrTr}$, $d^*_{TrS}$ denote the distributions of nearest-neighbor distances from train to train and train to synthetic, respectively. The x- and y-axes represent the distances and the eCDF, respectively, and are both shown on a $\log_{10}$ scale. The eCDF of $d^*_{TrS}$ is underfit for RBM1 and RBM3, well aligned with $d^*_{TrTr}$ for RBM7—indicating no expected anomaly—and slightly overfit for RBM12.

## G PRIVET ALGORITHM

---

**Algorithm 1:** PRIVET

---

**Input**  : Tr $\in \mathbb{R}^{N_{Tr} \times D}$ (Train), Te $\in \mathbb{R}^{N_{Te} \times D}$ (Test), S $\in \mathbb{R}^{N_S \times D}$ (Synthetic)

**Output** : $\Delta \pi_r, \mathbf{1}_{\{\Delta \pi_r \leq Th\}} \quad \forall r \in \{1, ..., N_S\}$

1) Compute 1-NN distances:
- $d^*_{TrTr}$ // nearest neighbor distances from train to train
- $d^*_{STr}$ // nearest neighbor distances from synthetic to train
- $d^*_{STe}$ // nearest neighbor distances from synthetic to test

2) Sort $d^*_{TrTr}, d^*_{STr}, d^*_{STe}$ with all distances $\delta^{STr}_r \in d^*_{STr}$ (respect. $\delta^{STe}_r \in d^*_{STe}$) in increasing order indexed by their rank $r$.

3) Fit EVD on $d^*_{TrTr}$ i.e. find estimators $\hat{A}, \hat{\alpha}$ or $\hat{A}, \hat{B}$ in (1) for $\hat{G}_{d^*_{TrTr}}$

**for** $r \in \{1, ..., N_S\}$ **do**

$$\pi^{Tr}_r = 1 - \sum_{q=0}^{r} \binom{N_S}{q} \left[ \hat{G}_{d^*_{TrTr}}(\delta^{STr}_r) \right]^q \left[ 1 - \hat{G}_{d^*_{TrTr}}(\delta^{STr}_r) \right]^{N_S - q};$$

$$\pi^{Te}_r = 1 - \sum_{q=0}^{r} \binom{N_S}{q} \left[ \hat{G}_{d^*_{TrTr}}(\delta^{STe}_r) \right]^q \left[ 1 - \hat{G}_{d^*_{TrTr}}(\delta^{STe}_r) \right]^{N_S - q};$$

$$\Delta \pi_r = \log \frac{\pi^{Tr}_r}{\pi^{Te}_r};$$

**end**

---

## H IMAGE DATA

For the copycat experiment (c.f. Main.Section. 3.2), we used PyTorch's CIFAR10 train and test. We used synthetic data generated by SOTA diffusion model on CIFAR10 Xu et al. (2023) that was provided by Stein et al. (2023) and down-sampled it to 10,000 images instead of the 100,000 available. The down-sampling was achieved by taking the first 1,000 images of each class. The pseudo-synthetic data is further created by selecting the first $\lfloor \beta N \rfloor$ training examples and the first $N - \lfloor \beta N \rfloor$ synthetic examples (where $N$ is the size of the synthetic set), thereby deterministically injecting a fraction $\beta$ of real samples into the synthetic distribution.

For the computer vision transform, we used the following:

- `Posterize(bits=5)`: Applies bit-level quantization by reducing each color channel to 5 bits using `torchvision.transforms.functional.posterize`, simulating color degradation.

- `CenterCrop(28) + Pad(2)`: Crops the center 28×28 region of the input image and pads it by 2 pixels on each side, preserving input shape and emphasizing central image content.

- `JPEGQuality(quality=75)`: Re-encodes a `PIL.Image` using JPEG compression at quality level 75 via an in-memory `BytesIO` buffer, simulating lossy compression; used within a `torchvision.transforms.Compose` pipeline.

- `ElasticTransform()`: Applies elastic deformation to the image using `torchvision.transforms.ElasticTransform` with default parameters, introducing spatial distortions that simulate realistic image warping.

`fld.features.DINOv2FeatureExtractor` from FLD github was used to extract DINOv2 embeddings. Default value corresponds to ViT-B/14 distilled without registers. In practice we observed similar

behavior for embeddings produced by ViT-B/14 and ViT-L/14 models, we thus used the former for computational efficiency reason.

For PRIVET, we fit the first $0.1\%$ to $2\%$ of the NN euclidean distances between train-train DINOv2 embeddings with a Weibull distribution. This corresponds to the first mode observed on the CDF. We used a renormalization factor equal to $5^{\frac{1}{25}}$ for rescaling the distances in the distribution of NN from synthetic-to-test samples ($N_{train} = 5N_{test}$).

For PQMass, we set `re_tesselation = 1000`, following recommendations that higher values yield improved performance.

$\beta$ fraction of copied training samples are in $\{0, 0.001, 0.01, 0.1, 0.2, 0.3, 0.4, 0.5, 0.6, 0.7, 0.8, 0.9, 1\}\%$

For wavelet embeddings we used `wavelet = "sym5"` with `max_level = 3`, resulting in feature vectors of dimension 23,232 for CIFAR-10 images with shape $(3, 32, 32)$ (channels, height, width).

The limited performance of PRIVET (NPL) in detecting privacy leaks in certain cases (see Main Fig.3) can be better understood by examining the eCDFs shown in Fig.14. In these instances, the synthetic-to-train eCDFs appear underfit, even though separation of synthetic-to-train and synthetic-to-test eCDFs indicates privacy leakage that should be detected by $\overline{\Delta\pi_r}$ and the other scores described in Appendix B.

**Settings for benchmarked metrics** The settings were set similarly in the image and the genetic controlled experiments. Details are described in section E.4.

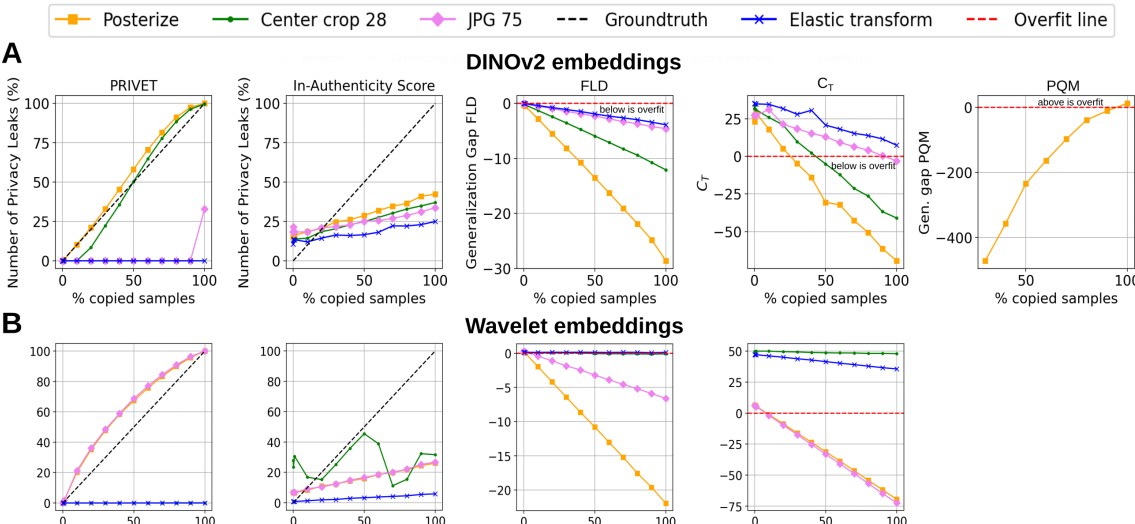

Figure 13: **Copycat experiment.** Several computer vision transformations (posterize, center crop 28, JPG 75, Elastic transform) are applied on CIFAR10 synthetic and training samples to generate a pseudo-synthetic dataset. x-axis represents the fraction of synthetic samples replaced with training ones; y-axis reports the value of the corresponding metric. For PRIVET and In-Authenticity, we report the percentage of identified privacy leaks. In the case of PRIVET, we use NPL from Eqs. 3. For Gen. gap (FLD) and $C_T$, overfitting is detected when the metric falls below the horizontal dashed line, whereas for Gen. gap (PQMass), overfitting is indicated when the metric exceeds the threshold. The top row presents results using DINOv2 (ViT-B/14 distilled without registers) embeddings of CIFAR10, while the bottom row shows results using wavelet packet coefficient space (symlet5, level=3). PQMass results are omitted for several transformations due to divergence, and are not shown for wavelet embeddings due to the high dimensionality, which caused excessive CPU time or memory limitations on GPU.

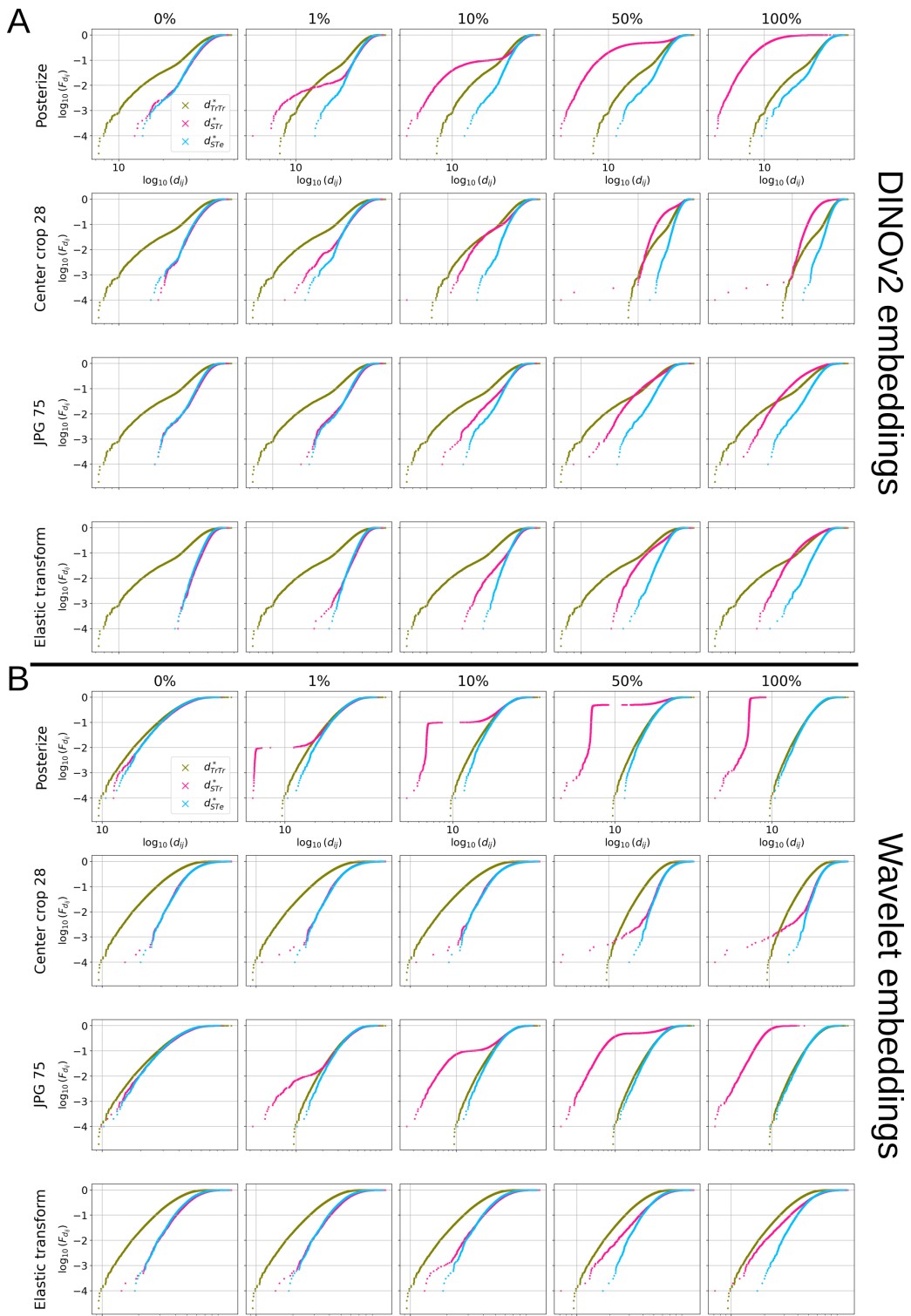

Figure 14: **eCDFs for copycat experiment.** Rows are posterize, center crop 28, JPG75 and elastic transform respectively. Columns are fraction of copied samples from train into synthetic set (transformed beforehand) corresponding to $0, 1, 10, 50, 100\%$ respectively. $d^*_{TrTr}$, $d^*_{STr}$, and $d^*_{STe}$ denote the distributions of nearest-neighbor distances from train to train, synthetic to train, and synthetic to test, respectively. The x- and y-axes represent the distances and the eCDF, respectively, and are both shown on a $\log_{10}$ scale. **A.** eCDFs using DINOv2 embeddings. **B.** eCDFs using wavelet features.

# I  MEMBERSHIP ATTACK ECDFS

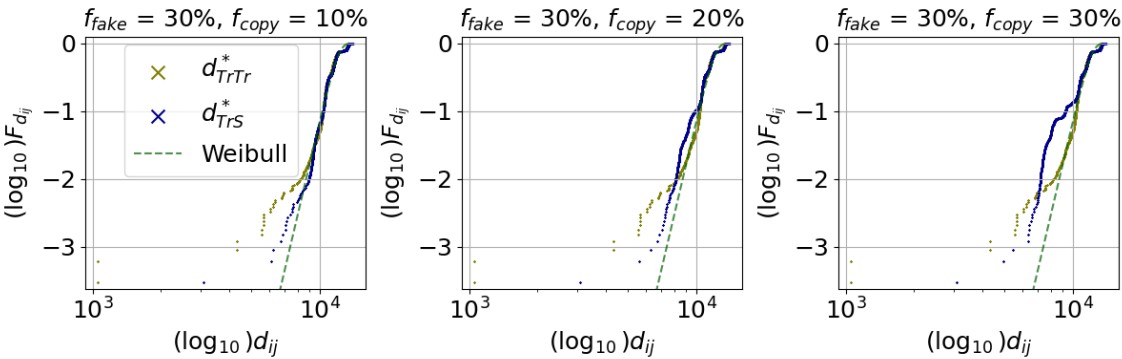

Figure 15: **eCDFs on different configution of leakage on 65,535 SNPs and 1,668 samples.** $d^*_{TrTr}$, $d^*_{TrS}$ denote the distributions of nearest-neighbor distances from train to train and train to synthetic, respectively. The x- and y-axes represent the distances and the eCDF, respectively, and are both shown on a $\log_{10}$ scale. The fraction of leaked samples, $f_{\text{fake}}$, is set to 30%, while the fraction of copied SNPs, $f_{\text{copy}}$, is equal to 10, 20 and 30%, respectively.

# J  COMPUTATIONAL EFFICIENCY COMPARISON

A single A100-48GB was used throughout all this study. We report the runtime of each algorithm for each transform and each fraction of copied samples in the copycat experiment. For PRIVET, the complexity is dominated by computing the 1-NN distances between synthetic-to-train and synthetic-to-test samples; the EVD fit itself is essentially instantaneous, and computing the $N_{train}$ 1-NN distances within the training set takes less than a minute.

Table 3: Computational efficiency comparison (time in seconds, mean ± std) for the copycat experiment on CIFAR10 ($N_{train} = 50,000, N_{synth} = 10,000, N_{test} = 10,000$).

| Method | DINOv2 embeddings (768-dim) | Wavelet embeddings (23,232-dim) |
|---|---|---|
| PRIVET | $9.97 \pm 0.10$ | $10.69 \pm 0.17$ |
| FLD | $1.01 \pm 0.09$ | $5.62 \pm 0.09$ |
| AUTH | $0.04 \pm 0.01$ | $1.74 \pm 0.02$ |
| CT | $8.38 \pm 0.60$ | $105.34 \pm 1.11$ |
| PQM (CPU) | $333.33 \pm 9.61$ | — |

We observed that PQM runs significantly faster on GPU compared to CPU. However, due to memory constraints, it could not be applied to wavelet embeddings, and was not evaluated on DINOv2 embeddings in the current study.

Further speedups are readily obtainable by relying on FAISS library Johnson et al. (2021); Douze et al. (2025) which provides exact and approximate nearest-neighbor search. This represents a straightforward avenue

for additional scalability improvements, but need to be tested when exact procedures are replaced by fast approximate ones.

## K  DIFFERENTIAL PRIVACY

For the diffusion baseline, we train a DDPM on a small subset of CIFAR-10. We first construct a deterministic train/validation split by sampling 1,000 training and 1,000 validation images from the CIFAR-10 training set with a fixed PyTorch generator seed (`GLOBAL_SEED = 43`). The denoising network is a UNet from the `diffusers` library (`UNet2DModel`) configured for $32 \times 32$ RGB images with channel widths $[64, 128, 256, 512]$, two residual blocks per scale, and self-attention at the deeper layers. The diffusion process uses $T = 1000$ timesteps with a cosine noise schedule: we interpolate $\beta_t$ between $10^{-4}$ and $2 \cdot 10^{-2}$ on a cosine grid. Training follows the standard noise-prediction objective. We optimize using AdamW with learning rate $2 \cdot 10^{-4}$, no weight decay, batch size 32, and train for 50 epochs. All pseudo-random generators (Python, NumPy, PyTorch CPU/GPU) are seeded, and cuDNN is set to deterministic mode.

To ensure that changes in privacy noise are the only source of stochasticity in our experiments, we fix the entire sampling pipeline. We first draw a bank of $N = 1000$ terminal noise samples $z_T^{(i)} \sim \mathcal{N}(0, I)$ with a fixed GPU generator seed, and a "reverse noise bank", a tensor of size $T \times N \times 3 \times 32 \times 32$ containing all Gaussian noises used in the reverse diffusion steps, also sampled once with a fixed seed. At test time we use the closed-form DDPM posterior $p_\theta(x_{t-1} \mid x_t)$ from Ho et al. (2020), i.e.,

$$\mu_t(x_t, \varepsilon_\theta) \; = \; \frac{1}{\sqrt{\alpha_t}}\left(x_t - \frac{\beta_t}{\sqrt{1-\bar{\alpha}_t}}\varepsilon_\theta(x_t, t)\right), \qquad \tilde{\sigma}_t^2 \; = \; \beta_t \frac{1-\bar{\alpha}_{t-1}}{1-\bar{\alpha}_t},$$

and set

$$x_{t-1} = \begin{cases} \mu_t(x_t, \varepsilon_\theta) + \tilde{\sigma}_t \, \xi_{t-1}^{(i)} & \text{if } t > 1, \\ \mu_t(x_t, \varepsilon_\theta) & \text{if } t = 1, \end{cases}$$

where the $\xi_{t-1}^{(i)}$ are taken deterministically from the precomputed noise bank. Given fixed trained weights, the synthetic dataset of 1000 samples produced by this procedure is therefore deterministic.

The quantile range for fitting PRIVET on train-train 1NN distances is $[1\%, 20\%]$, and the score is the number of privacy leaks in the underfitting regime ($n_{\text{pleaks}}(r)$, Eq. 9).

For the DP experiments, we reuse the exact same data split, UNet architecture, training hyperparameters, and deterministic sampling pipeline as in the baseline. The only change is that training is performed with differentially private SGD using Opacus. We sweep over noise multipliers

$$\sigma \in \{5.0, \, 3.0, \, 2.5, \, 2.0, \, 1.5, \, 1.0, \, 0.7, \, 0.5, \, 0.4, \, 0.3\},$$

and set the per-sample clipping threshold to a fixed `max_grad_norm` $= C$ (here $C = 10$). We use the PRV accountant implemented in Opacus and set the privacy failure probability to

$$\delta = \tfrac{1}{N_{\text{train}}} = 10^{-3}.$$

For each noise multiplier we train for 50 epochs and report the resulting $\varepsilon(\delta)$. After training, we generate exactly the same 1,000 synthetic samples as in the baseline run. Because all other sources of randomness (data split, model initialization, sampling noise, and embedding computation) are held fixed by seeding, differences in NPL across $\varepsilon$ can be attributed solely to the additional DP noise injected into the gradients during training.

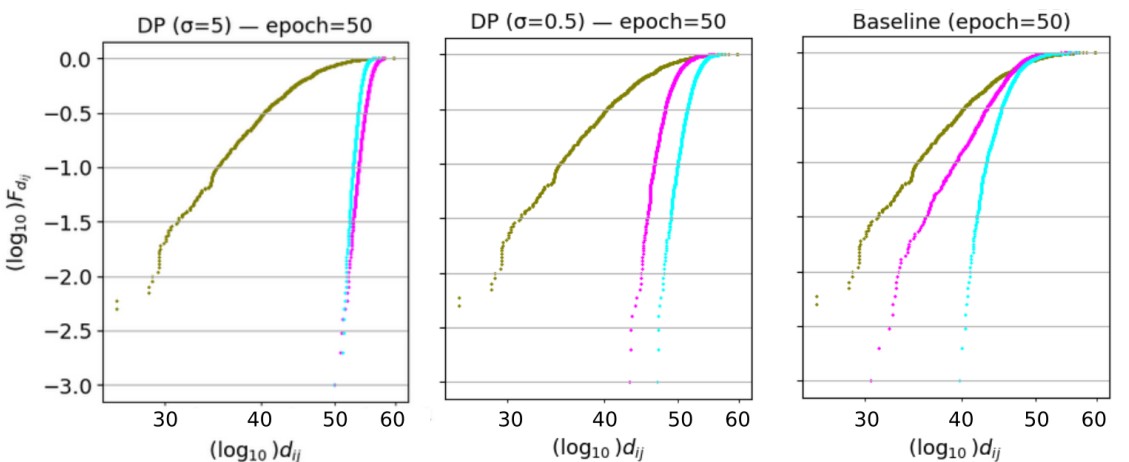

Figure 16: **Evolution of the eCDFs during training of DP-DDPM for representative privacy levels**. Green, magenta, and cyan curves correspond respectively to train-train ($d^*_{TrTr}$), synthetic-train ($d^*_{STr}$), and synthetic-test ($d^*_{STe}$) 1-NN distances. The x- and y-axes represent the distances and the eCDF, respectively, and are both shown on a $\log_{10}$ scale.

## L  IMPACT OF EMBEDDING ON THE CHARACTERIZATION OF UNDERFITTING/OVERFITTING REGIME

Fig. 17 highlights synthetic samples generated by the PFGMPP diffusion model that are considered as underfitting when evaluated with DINOv2 embeddings. In particular, examining the tail of the $d^*_{STr}$ distribution, we find synthetic samples that are visually nearly identical to their nearest neighbors in the training set, yet their embedding distances are almost twice as large as the distances between that training sample and its 1-NN in the training set (which is a duplicate). A similar case is shown in the second row. In contrast, the third row presents a pair of clearly different images ; the distance of the (synthetic, real) pair (bottom right) is comparable to that of the nearly identical pairs of the two previous rows, further illustrating the limitations of the embedding when identifying near-duplicate samples. This, together with Main Fig. 3, confirm the impact of embedding choice on all distance-based metrics and on the characterization of underfitting and overfitting samples.

## M  BROADER IMPACTS

The growing use of generative AI for synthetic data generation raises serious privacy concerns, particularly when models are trained on sensitive data such as copyrighted content or human genetic information governed by privacy regulations. Although deep generative modeling has seen rapid progress, the development of rigorous evaluation methods, especially for privacy auditing, has not kept pace. Existing privacy assessments typically rely on global, dataset-level metrics that offer limited interpretability and overlook individual-level risks. In this work, we address this gap by introducing a sample-level privacy metric capable of detecting overfitting and privacy leaks in synthetic datasets. Our method, PRIVET, provides interpretable, fine-grained privacy scores and is applicable across data modalities. We believe this contributes a necessary layer of

accountability and transparency to the deployment of generative models, particularly in high-stakes domains like biomedical research and content-sensitive applications.

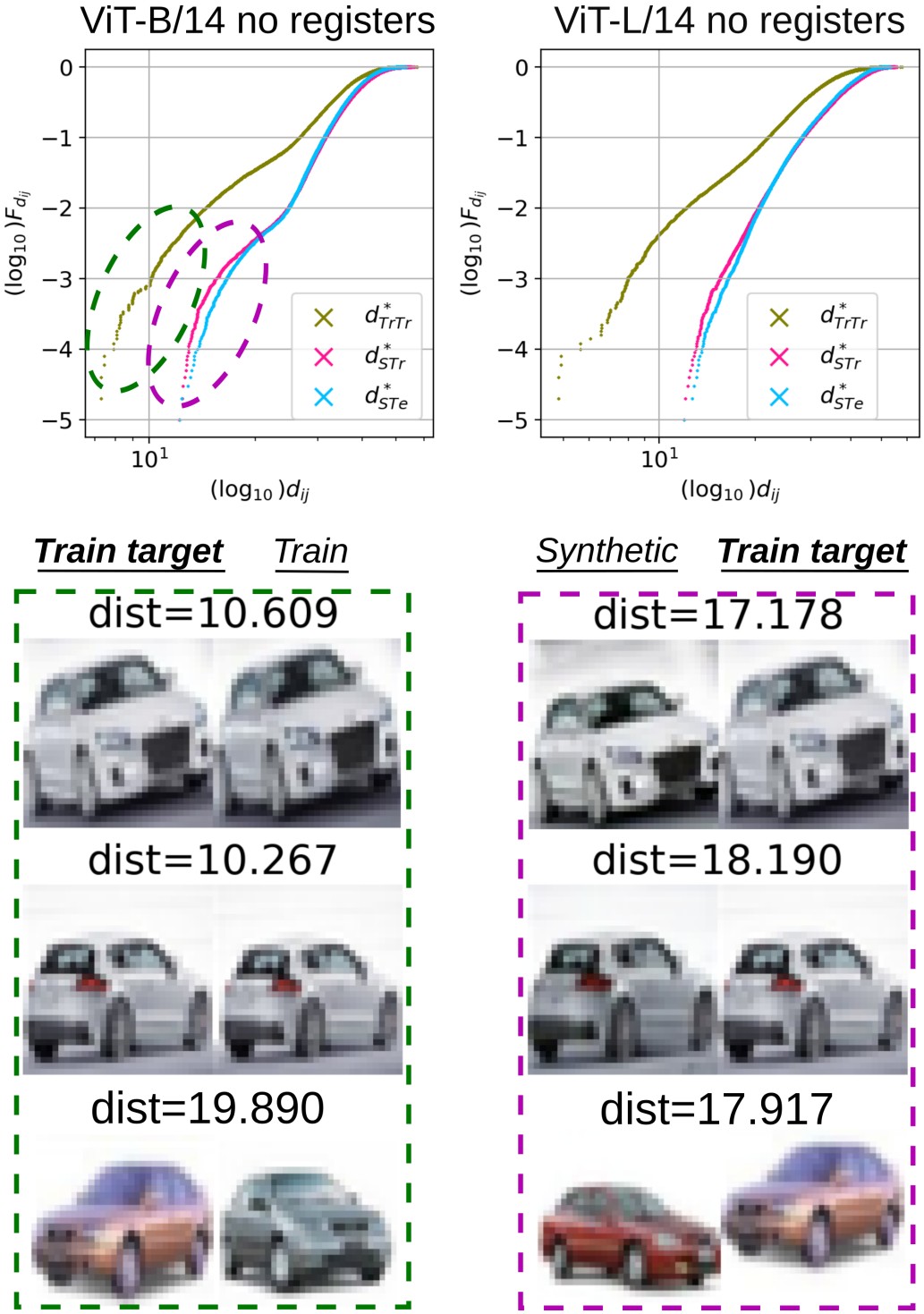

Figure 17: **Synthetic samples generated by PFGMPP appear underfit when using DINOv2 embeddings.** Synthetic samples are generated by PFGMPP diffusion model. **Top.** Distances are computed on DINOv2 embeddings using ViT-B/14 (86M parameters) without registers (left) and ViT-L/14 (300M parameters) without registers (right). $d_{TrTr}^*$, $d_{STr}^*$, and $d_{STe}^*$ denote the distributions of nearest-neighbor distances from train to train, synthetic to train, and synthetic to test, respectively. **Bottom.** We exhibit three pairs of train-train (left column) and synthetic-train (right column), where one training sample is common in both pairs (train target). These pairs appear in the tail of the eCDFs (encircled in color). The distance is the euclidean distance on DINOv2 embeddings.