# OpenReview forum: "PRIVET: PRIVacy metric based on Extreme value Theory"
_ICLR.cc/2026/Conference — Submitted to ICLR 2026_

### Official Review · Reviewer_6mFF · 2025-10-30

**Soundness:** 3
**Presentation:** 3
**Contribution:** 3
**Rating:** 6
**Confidence:** 2

**Summary:**

This paper presents PRIVET, a sample-level privacy metric for synthetic data that leverages Extreme Value Theory (EVT) applied to nearest-neighbor (NN) distances. The method fits a statistical model (Weibull or Gumbel) to characterize how close synthetic samples are to real training data, identifying those that are unusually similar and may indicate memorization or leakage. PRIVET provides both individual privacy scores and aggregate dataset indices, offering interpretable, quantitative assessments of privacy risk. The approach is modality-agnostic and is validated on genomic and image datasets, where it consistently detects privacy leakage more effectively than existing global-only metrics.

**Strengths:**

1. Provides sample-level privacy scores rather than global aggregates.
2. Grounded in Extreme Value Theory (EVT), offering statistical interpretability.
3. Scalable and modality-agnostic across images and genomics.
4. Demonstrates strong empirical validation and thoughtful limitations.

**Weaknesses:**

1.  Embedding dependence: PRIVET’s results in vision tasks vary substantially across different embeddings (e.g., DINOv2 vs. wavelets). Since the metric depends on nearest-neighbor distances, poor or mismatched embeddings can distort similarity relationships and lead to inconsistent privacy estimates. A systematic embedding sensitivity analysis or adaptive embedding selection strategy would strengthen the method’s robustness.
2. Threshold and tail calibration: The choice of the EVT fitting region and the leak-flagging threshold (τ) are largely heuristic. These parameters influence both the sample-level scores and the number of privacy leaks (NPL), which may not be consistent across datasets. Providing a calibration method (e.g., via null distributions or false-positive control) or ablations on these parameters would clarify interpretability.
3. Handling of structured or heterogeneous data types: While the method is demonstrated on genomics (Hamming distances) and images (embedding distances), many real-world synthetic data scenarios involve mixed types (categorical, continuous, time series, hierarchical). It is unclear how PRIVET’s nearest-neighbor + EVT framework adapts to such heterogeneous or relational data. The paper could benefit from discussing or testing on more complex data types beyond the two modalities shown.

**Questions:**

1. Parameter sensitivity: How sensitive are PRIVET’s results to the choice of EVT fitting range and the leak-detection threshold (τ)? Have you explored data-driven calibration methods or null-distribution bootstrapping to make  τ more interpretable?

2. Embedding robustness: Since performance varies across embeddings (especially in vision), have you considered adaptive or learned representations that better preserve privacy-relevant distances? Could embedding selection be automated based on EVT goodness-of-fit or validation metrics?

3.  Generality across data types: PRIVET is shown on genetic and image data, two well-structured modalities. How would the approach generalize to mixed or structured data (e.g., tabular datasets with categorical and continuous variables, or time-series data)? Are there distance metrics or representations that could preserve EVT assumptions in such cases?

---

> ### Author Response · Authors · 2025-11-21
>
> > Embedding dependence: PRIVET’s results in vision tasks vary substantially across different embeddings (e.g., DINOv2 vs. wavelets).
>
> The method itself is formulated in a fully general way and does not impose any assumptions on the data modality. In practice, some data types benefit from an embedding space while others do not. The choice of representation is left to the practitioner, who should select the space that best captures the relevant structure of the data. This flexibility is a feature rather than a limitation of the approach, in our viewpoint. Moreover, PRIVET provides multiple scores tailored to different regimes (overfitting, and leakage in underfitting). Using the score appropriate for the scenario at hand substantially reduces dependence on the specific embedding. As shown in the revised results (e.g., Fig. 14 in Appendix L), once the proper score is applied, PRIVET should remain sensitive across embeddings. Considering the reiterate remarks of different reviewers about this point we plan to include a discussion about this in the main text.
>
> > Since the metric depends on nearest-neighbor distances, poor or mismatched embeddings can distort similarity relationships and lead to inconsistent privacy estimates. A systematic embedding sensitivity analysis or adaptive embedding selection strategy would strengthen the method’s robustness.
>
> We agree that embeddings can meaningfully influence nearest-neighbor structure, and part of our contribution is precisely to highlight these issues rather than assume that existing embeddings are always appropriate. A systematic embedding sensitivity analysis or an adaptive embedding selection mechanism would indeed strengthen robustness, however, developing such methods lies outside the scope of this work.
>
> > Threshold and tail calibration: The choice of the EVT fitting region and the leak-flagging threshold ($\tau$) are largely heuristic. These parameters influence both the sample-level scores and the number of privacy leaks (NPL), which may not be consistent across datasets. Providing a calibration method (e.g., via null distributions or false-positive control) or ablations on these parameters would clarify interpretability.
>
> We thank the reviewer for raising this important point. Appendix D (Fig. 5) provides a meta-analysis across datasets and modalities that evaluates the quality of the EVT fit under variations of the fit’s hyperparameters. We agree that extending this analysis to the full pipeline up to the privacy score would provide additional insight. In the current work, the threshold was fixed to the default value $-3$, corresponding to flagging a rare event of probability $\frac{1}{1000}$. We agree that a systematic study of threshold calibration would be valuable. A natural approach would be to use a train–validation split to select a threshold $\tau$ along the Pareto front of the precision–recall curve, followed by evaluation on a held-out test set. We view this as an important direction for future work.
>
> > Handling of structured or heterogeneous data types: While the method is demonstrated on genomics (Hamming distances) and images (embedding distances), many real-world synthetic data scenarios involve mixed types (categorical, continuous, time series, hierarchical). It is unclear how PRIVET’s nearest-neighbor + EVT framework adapts to such heterogeneous or relational data. The paper could benefit from discussing or testing on more complex data types beyond the two modalities shown.
>
> We thank the reviewer for raising this point. As shown in Appendix Fig. 5, the first steps of the metric (computing 1NN distances and checking whether their distribution exhibits EVT behavior) were performed on heterogeneous tabular datasets such as UCI Bank and UCI Adult, which contain both categorical and continuous variables. These datasets were handled through standard preprocessing: continuous variables were standardized, categorical variables were one-hot encoded, and Euclidean distances were computed on the resulting representation. The corresponding figure shows that the 1NN distance distribution still follows the EVT regime under this mixed-data setup. Moreover, prior work such as Meeus et al., “Achilles’ Heels: Vulnerable Record Identification in Synthetic Data Publishing,” ESORICS 2023, proposes alternative distance functions (e.g., their Equation 5) specifically designed for heterogeneous data.
>
> > Parameter sensitivity: How sensitive are PRIVET’s results to the choice of EVT fitting range and the leak-detection threshold ($\tau$)? Have you explored data-driven calibration methods or null-distribution bootstrapping to make $\tau$ more interpretable?
>
> Please see the goodness-of-fit analysis in Appendix D and our detailed response above. Appendix D evaluates EVT fit quality under hyperparameter variations, and we outline how threshold calibration could be performed using validation-based selection.

---

> ### Author Response · Authors · 2025-11-21
>
> > Embedding robustness: Since performance varies across embeddings (especially in vision), have you considered adaptive or learned representations that better preserve privacy-relevant distances?
>
> As discussed above, performance differences across embeddings can be addressed through appropriate score selection. Adaptive or learned representations are an interesting direction but beyond the scope of the present work.
>
> > Could embedding selection be automated based on EVT goodness-of-fit or validation metrics?
>
> This is indeed an interesting idea. In principle, embedding choice could be guided by EVT goodness-of-fit or validation-based criteria. In practice, however, implementing such a procedure at a general level is impractical: it would require computing or maintaining multiple embeddings, and the right embedding family depends heavily on the specific domain and data modality. That said, for a given application, users could certainly adopt such a strategy.
>
> > Generality across data types: PRIVET is shown on genetic and image data, two well-structured modalities. How would the approach generalize to mixed or structured data (e.g., tabular datasets with categorical and continuous variables, or time-series data)? Are there distance metrics or representations that could preserve EVT assumptions in such cases?
>
> Please see our answer above. Mixed tabular datasets (categorical + continuous) were already evaluated in Appendix Fig. 5. EVT behavior was preserved via standard preprocessing, and other distances for heterogeneous data exist in the literature (e.g., Meeus et al., ESORICS 2023). We plan to insert this discussion in the main text.

---

> > ### Comment · Reviewer_6mFF · 2025-11-25
> >
> > Thank you for the clarifications.

---

### Official Review · Reviewer_L37y · 2025-11-01

**Soundness:** 3
**Presentation:** 3
**Contribution:** 3
**Rating:** 4
**Confidence:** 3

**Summary:**

This paper introduces PRIVET, a privacy evaluation metric for generative models that relies on extreme value theory to model the tail distribution of nearest-neighbor distances.
By comparing the NN distance distributions between synthetic, training, and test data, PRIVET provides both dataset-level and sample-level privacy assessments, capable of identifying overfitting and privacy leakage.
The authors claim PRIVET to be interpretable, scalable, and domain-agnostic, and provide experiments on genetic data and image data with comparisons to prior metrics such as AATS, CT, FLD, PQMass, and AUTH.

**Strengths:**

1. The paper applies extreme value theory to model the nearest-neighbor distance tails, providing a statistically grounded method for privacy leakage detection.

2. PRIVET can generate interpretable, per-sample privacy scores while remaining consistent with dataset-level statistics, which bridges the gap between global metrics (like AATS, FLD) and local boolean methods (like AUTH).

3. Experiments on both genetic and image datasets demonstrate the versatility of the method, showing good scalability to high-dimensional data and small-sample regimes.

**Weaknesses:**

1. Table 1’s distinction between “interpretable” and “non-interpretable” metrics is not clearly defined. It appears that *interpretability* is equated with having a boolean “leak / not leak” label, but the table also lists a “real value” column without clear justification. Moreover, the rationale for classifying methods such as AUTH as interpretable only at the sample level, but not at the dataset level, is insufficiently explained.
2. The paper claims that prior metrics like FLD and PQMass fail under low-data conditions or produce noisy estimates, yet this claim is not supported by empirical evidence.
3. While the experiments cover genetic (binary) and image (continuous) data, the claim of being “domain-agnostic” is incomplete as it lacks a textual or sequential modality.
4. The evaluation focuses mainly on precision and recall.

**Questions:**

1. Can you provide quantitative or visual evidence supporting the claim that previous metrics (e.g., FLD, PQMass) fail or become noisy in low-data regimes?

2. Since experiments only include genetic and image data, could you discuss or show results on text data to better support the “domain-cross” generalization claim?

3. Would it be possible to add metrics such as ROC-AUC or F1-score to strengthen the evaluation?

---

> ### Author Response · Authors · 2025-11-21
>
> > Table 1’s distinction between “interpretable” and “non-interpretable” metrics is not clearly defined. It appears that interpretability is equated with having a boolean “leak / not leak” label, but the table also lists a “real value” column without clear justification.
>
> We appreciate the reviewer’s request for clarification. In our terminology, a metric is considered non-interpretable when it outputs a continuous, unbounded score that provides only a general indication of how severe the privacy leakage is, and does not identify which samples are affected. By contrast, we classify metrics as interpretable when their outputs correspond to a probability of leakage or a boolean indicator of whether leakage has occurred, on a per-sample basis. We will revise the text to make this definition more explicit.
>
> > Moreover, the rationale for classifying methods such as AUTH as interpretable only at the sample level, but not at the dataset level, is insufficiently explained.
>
> We believe there is a misunderstanding here, as in Table 1, AUTH is indeed classified as interpretable at the dataset level since the quantity refers to the fraction of leaked samples.
>
> > The paper claims that prior metrics like FLD and PQMass fail under low-data conditions or produce noisy estimates, yet this claim is not supported by empirical evidence.
>
> Empirical evidence for this claim is provided in Appendix E.1 (Fig. 6), where we explicitly evaluate the low–data regime on genetic data. In this setting, both FLD and PQMass failed to run, each method crashed due to insufficient sample size, while only PRIVET, AATS and AUTH remained operational.
>
> > While the experiments cover genetic (binary) and image (continuous) data, the claim of being “domain-agnostic” is incomplete as it lacks a textual or sequential modality.
>
> Genetic data are inherently sequential, and thus already provide coverage of a sequential modality. Extending the evaluation to text or other sequence types is certainly possible, but the notion of privacy leakage in linguistic data is not yet well established in the literature, which complicates meaningful evaluation. We note, however, that we also evaluated tabular datasets with heterogeneous data types for goodness-of-fit, there, the distribution of distances also exhibits clear asymptotic behavior consistent with the EVT regime.
>
> > The evaluation focuses mainly on precision and recall.
>
> We would like to clarify that focusing on precision and recall was an evaluation choice intended to convey the most relevant information within the limited space of the paper. Since the F1 score is essentially a summary of precision and recall, we judged that it would not add substantial insight in the main text. Nonetheless, our evaluation is not restricted to these metrics: Appendix Figures 7 and 8 provide a full decomposition into TP, FP, TN, and FN, and F1 scores are reported in Appendix E.5 (Fig. 10). ROC–AUC values can also be provided. We do not include ROC curves because each controlled-leakage experiment produces a grid of scenarios, each with its own intrinsic ROC curve, making it impractical to visualize all of them on a single plot.
>
> > Can you provide quantitative or visual evidence supporting the claim that previous metrics (e.g., FLD, PQMass) fail or become noisy in low-data regimes?
>
> Please see our detailed response above. In Appendix E.1 (Fig. 6), we provide empirical evidence showing that FLD and PQMass fail to run in the low-data genetic setting due to insufficient sample size, whereas PRIVET remains operational.
>
> > Since experiments only include genetic and image data, could you discuss or show results on text data to better support the “domain-cross” generalization claim?
>
> As discussed above, genetic data are already sequential, and the notion of privacy leakage for linguistic data is not yet well defined in the literature.
>
> > Would it be possible to add metrics such as ROC-AUC or F1-score to strengthen the evaluation?
>
> As indicated above, F1 scores are already reported in Appendix E.5 (Fig. 10), and ROC–AUC values can also be provided. ROC curves are not shown because each controlled leakage experiment generates a grid of scenarios, each with its own ROC curve, making combined visualization impractical.

---

### Official Review · Reviewer_xbm4 · 2025-11-02

**Soundness:** 2
**Presentation:** 2
**Contribution:** 2
**Rating:** 4
**Confidence:** 2

**Summary:**

The paper proposes PRIVET, a privacy audit for generative models that assigns a sample-level privacy-leak score using extreme value theory (EVT) fitted to the lower tail of nearest-neighbor (NN) distance distributions. The method models the empirical CDF of train–train 1-NN distances with a Weibull or Gumbel distribution, then evaluates where each synthetic sample's NN distance to train or test falls within the fitted tail via order-statistics. It reports both local scores and global indices, including the average log-ratio between train-referenced and test-referenced probabilities and a count of suspected leaks under a threshold. Experiments cover controlled genetic-sequence settings with configurable leak rate and copied-bits fraction, a CIFAR-10 "copycat" setup using DINOv2 and wavelet embeddings, and a membership-inference style analysis on RBM-generated genomic data. The authors argue PRIVET is interpretable, scalable, domain-agnostic, and more sensitive than baselines such as AATS, CT, FLD, PQMass, and the Authenticity score in several regimes.

**Strengths:**

1) Broad empirical coverage. The study spans high-dimensional genetics, computer vision with modern embeddings, and a membership-attack scenario, which together demonstrate versatility across modalities and training regimes, including underfitting.
2) Interpretability and actionable outputs. The ability to flag specific synthetic samples as probable leaks and to summarize with an estimated number of leaks is useful in governance workflows. The method surfaces when embedding choices compromise detectability.
3) Comparison to multiple baselines. The paper contrasts PRIVET with several recent metrics and highlights conditions where others become noisy, require matched sizes, or provide only global signals.

**Weaknesses:**

1) Embedding dependence undermines "modality-agnostic" positioning. Results show that detectability varies substantially with the representation, with failures for certain image transforms under DINOv2 and improvements with wavelets. The method is only as good as the embedding and distance, which weakens the claim of generality. A guidance section on selecting or learning privacy-aware embeddings, or an adaptive metric learning step, would be valuable.
2) Figure~1 readability. The numerical annotations in Figure~1 overlap, particularly within dense regions of the scatter, which reduces interpretability and obscures the intended comparison between train-referenced and test-referenced EVT fits. Clear labeling is crucial for a method whose main selling point is interpretability.
3) Image experiments underperform relative to In-Authenticity in certain regimes} In the CIFAR-10 evaluations (Figure~3), PRIVET does not consistently outperform the In-Authenticity baseline. In some transformations such as JPEG compression and elastic distortion, PRIVET's separation between memorized and novel samples appears weaker. These results directly challenge the general claim that the proposed approach is "more sensitive" across modalities.
4) Lack of empirical scalability evaluation. The paper asserts that PRIVET is scalable because it relies on nearest-neighbor distances and closed-form EVT fits. However, there is no time or memory comparison against competing privacy auditing methods such as AATS, CT, FLD, or PQMass. The experiments do not report runtime on increasing dataset sizes, nor the computational effect of high-dimensional embeddings or approximate nearest-neighbor search. Since practitioners must often audit millions of samples in regulatory contexts, an explicit complexity analysis and empirical runtime study are necessary to substantiate the scalability claim.

**Questions:**

1) Can you revise Figure~1 to avoid overlapping numerical labels and provide clearer visual separation of extreme-tail points? Since interpretability is a core claim, visual clarity is essential for evaluating sample-level privacy scores.
2) In Figure~3, PRIVET performs worse than In-Authenticity for certain image transformations such as JPEG compression and elastic distortions. What underlying factors drive these failures?
3) The method's success appears strongly influenced by representation choice (e.g., DINOv2 vs wavelets). Do you have recommendations or automatic adaptation mechanisms to ensure reliable performance across modalities without manual embedding engineering?
4) You assert PRIVET is scalable, yet runtime and memory comparisons are not reported. How does performance scale relative to baselines as dataset size and embedding dimensionality increase, especially under approximate nearest-neighbor search? Please include empirical runtime curves and complexity analysis.

---

> ### Author Response · Authors · 2025-11-21
>
> > Broad empirical coverage.
>
> Thank you for this kind comment.
>
> > Embedding dependence undermines "modality-agnostic" positioning.
>
> We respectfully disagree with this assessment. The method itself is formulated in a fully general way and does not impose any assumptions on the data modality. In practice, some data types benefit from an embedding space while others do not. The choice of representation is left to the practitioner, who should select the space that best captures the relevant structure of the data. This flexibility is a feature rather than a limitation of the approach, in our viewpoint. Furthermore, using the score appropriate for the scenario at hand substantially reduces dependence on the specific embedding. As shown in the revised results (e.g., Fig. 14 in Appendix L), once the proper score is applied, PRIVET should remain sensitive across embeddings. Considering the reiterate remarks of different reviewers about this point we plan to include a discussion about this point in the main text.
>
> > Results show that detectability varies substantially with the representation, with failures for certain image transforms under DINOv2 and improvements with wavelets.
>
> PRIVET provides multiple scores tailored to different regimes, such as underfitting and overfitting. An interesting avenue for improvement is to automate the selection of the most appropriate score for the situation at hand. Regarding the reviewer’s concern, we note that in Fig. 12 (Appendix), although certain embeddings make the synthetic data appear underfit, varying the leakage level still produces a clear divergence between the 1NN synthetic-to-train and synthetic-to-test curves across all transformations. This divergence is precisely what indicates privacy leakage, and PRIVET was designed in such a way that it is sensitive to it. In such cases, score (5) in Appendix B is the appropriate one to use. This aspect and alternatives scores could be upgraded in the main text.
>
> > The method is only as good as the embedding and distance, which weakens the claim of generality.
>
> Defining a meaningful distance on real datasets is in general a complex task, and this challenge is not specific to our method, it is a recurrent issue in generative model evaluation more broadly. This naturally puts more weight on the choice of distance. However, in continuous finite-dimensional spaces and in the limit of very large sample sizes, the method should work with essentially any reasonable distance, which makes it quite versatile. We actually discuss this point in the Limitations section, and Appendix C provides further details on distances in finite-dimensional continuous spaces.
>
> >  A guidance section on selecting or learning privacy-aware embeddings, or an adaptive metric learning step, would be valuable.
>
> We thank the reviewer for this valuable suggestion. For now, designing improved embeddings lies outside the scope of the present work. In practice, we recommend always inspecting the CDFs of the 1NN distances, as they provide immediate feedback on whether a given embedding is suitable
>
> > Figure1 readability. The numerical annotations in Figure1 overlap, particularly within dense regions of the scatter, which reduces interpretability and obscures the intended comparison between train-referenced and test-referenced EVT fits. Clear labeling is crucial for a method whose main selling point is interpretability.
>
> Figure 1 is intended as a schematic illustration of the different regimes: (i) underfitting, where the pink (1NN synthetic–train) and blue (1NN synthetic–test) curves lie below the green (1NN train–train) curve but remain aligned; (ii) underfitting with leakage, where pink and blue curves lie below the green curve but start to separate; (iii) realistic generation, where all curves align; and (iv) overfitting, where the blue and green curves align while the pink curve rises above them. Because this is a schematic figure rather than an actual plot, readability of the lower tail is not essential for interpreting the conceptual message.

---

> ### Author Response · Authors · 2025-11-21
>
> > Image experiments underperform relative to In-Authenticity in certain regimes. In the CIFAR-10 evaluations (Figure.3), PRIVET does not consistently outperform the In-Authenticity baseline. In some transformations such as JPEG compression and elastic distortion, PRIVET's separation between memorized and novel samples appears weaker. These results directly challenge the general claim that the proposed approach is "more sensitive" across modalities.
>
> As detailed in Appendix B, PRIVET can be instantiated through several scores, each suited to a specific regime (overfitting or privacy leakage within underfitting). In Figure 3, a single score was used uniformly across transformations, without tailoring it to the scenario at hand. This partly explains the weaker sensitivity observed for transformations such as JPEG compression or elastic distortions. When the appropriate score for the underfitting with leakage regime is used, specifically, score (5) in Appendix B, PRIVET successfully detects leakage. Figure 14 in Appendix L already illustrates this for elastic distortions. We are further providing results for this score across all transformations, and these will be included in the updated manuscript. With the correct score selection, PRIVET consistently shows higher sensitivity than competing metrics in this experiment. We plan to move these alternative scores to the main text.
>
> > Lack of empirical scalability evaluation. The paper asserts that PRIVET is scalable because it relies on nearest-neighbor distances and closed-form EVT fits. However, there is no time or memory comparison against competing privacy auditing methods such as AATS, CT, FLD, or PQMass. The experiments do not report runtime on increasing dataset sizes, nor the computational effect of high-dimensional embeddings or approximate nearest-neighbor search. Since practitioners must often audit millions of samples in regulatory contexts, an explicit complexity analysis and empirical runtime study are necessary to substantiate the scalability claim.
>
> A preliminary computational efficiency analysis is provided in Appendix J (Table 3), which shows that the metric exhibits reasonable computational behavior. The dominant cost lies in computing 1NN distances, where exact search scales quadratically with the number of samples. However, this is not prohibitive in practice when GPUs are available. For instance, as reported in Appendix 5, computation of distances and goodness-of-fit on a subset of 1M ImageNet samples was carried out and required under 5 minutes on an A100-40GB GPU. Importantly, PRIVET never stores the full pairwise distance matrix. Distances are computed in batches, so the memory footprint scales as $\text{batch}_X \times \text{batch}_Y$, with batch sizes much smaller than the dataset, leading to modest space complexity. Further speedups are readily obtainable by relying on FAISS library (Johnson et al., "Billion-scale similarity search with GPUs" IEEE T. Big Data, 2019; Douze et al., arXiv 2024) which provides exact and approximate nearest-neighbor search. This represents a straightforward avenue for additional scalability improvements, but need to be tested when exact procedures are replaced by fast approximate ones.
>
> > Can you revise Figure.1 to avoid overlapping numerical labels and provide clearer visual separation of extreme-tail points? Since interpretability is a core claim, visual clarity is essential for evaluating sample-level privacy scores.
>
> See answer above
>
> > In Figure.3, PRIVET performs worse than In-Authenticity for certain image transformations such as JPEG compression and elastic distortions. What underlying factors drive these failures?
>
> This has been discussed in the main text and we refers the reviewer to the corresponding figure in appendix, "This limitation can be attributed to DINOv2 embeddings, which cause synthetic samples to appear to be strongly underfitting (see Figure 12. Appendix.H for CDFs of 1-NN distances", and further discussed in Appendix K.
>
> > The method's success appears strongly influenced by representation choice (e.g., DINOv2 vs wavelets). Do you have recommendations or automatic adaptation mechanisms to ensure reliable performance across modalities without manual embedding engineering?
>
> The choice of representation can indeed influence performance. In practice, we recommend inspecting the CDFs of the 1NN distances, as they provide immediate feedback on whether the embedding yields meaningful nearest-neighbor structure. A suitable embedding is one in which the train–train distances form a coherent baseline and the synthetic–train and synthetic–test curves exhibit the expected relative ordering (e.g., underfitting, overfitting, privacy leakage in underfitting). This visual diagnostic helps ensure that PRIVET operates in a representation where its distance-based assumptions are valid.

---

### Official Review · Reviewer_9Y2e · 2025-11-11

**Soundness:** 2
**Presentation:** 3
**Contribution:** 2
**Rating:** 4
**Confidence:** 4

**Summary:**

The authors propose a new privacy metric based on the extension of the nearest neighborhood based statistics that have been used in the past. Simply, the proposed privacy score is defined based on r-th smallest nearest-neighbor (NN) distance to the reference dataset, computed from the M samples of the dataset of interest. In other words, for each synthetic sample, the proposed privacy metric takes the log ratio of overfitting scores between the training and test sets. According to authors, a low value indicates a statistical anomaly and may signal a potential privacy leak.

**Strengths:**

Another metric that addresses an important privacy problem.

Authors provide some interesting examples based on pseudo-synthetic data that suits their distance metrics and embeddings.

**Weaknesses:**

Only a single experiment using an RBM-generated synthetic dataset is presented, which limits the generality of the findings. Including additional datasets would strengthen the empirical validation.

Moreover, the current results do not clearly demonstrate whether the proposed approach detects any actual privacy risk. One potential improvement would be to employ a data generator with a controlled privacy parameter—for example, a differentially private generator with varying  epsilon values—and evaluate whether the proposed privacy score correlates with these privacy levels.

Finally, it would be valuable to extend the analysis beyond membership inference attacks. Incorporating sensitive attribute inference or other privacy attack scenarios could help assess whether the proposed score also correlates with stronger leakage in more complex attack models.

**Questions:**

None.

---

> ### Author Response · Authors · 2025-11-21
>
> > Only a single experiment using an RBM-generated synthetic dataset is presented, which limits the generality of the findings. Including additional datasets would strengthen the empirical validation.
>
> We thank the reviewer for raising this point. While it is true that we trained only a RBM ourselves, our evaluation is not limited to this single model: we also include experiments on synthetic data generated by a state-of-the-art diffusion model (PFGM++) trained on CIFAR-10, using the publicly released samples from Stein et al., “Exposing flaws of generative model evaluation metrics and their unfair treatment of diffusion models”, NeurIPS 2023. That said, we agree that expanding to additional datasets would further strengthen the empirical validation. Our intention in this submission is to provide a proof-of-concept study establishing the foundations of the metric. We also note that reviewer xbm4 considered the empirical coverage to be “broad”.
>
> > Moreover, the current results do not clearly demonstrate whether the proposed approach detects any actual privacy risk.
>
> We respectfully disagree with the reviewer. While discussed in the plain text maybe this was not completely clear. We follow the view point adopted in the AATS paper (Yale et al. "Generation and evaluation of privacy preserving synthetic health data" Neurocomputing 2020), which is that distortions of the NN distribution w.r.t. the train provides a measure of overfitting, while difference of distortion of the NN distribution w.r.t. the test compared to the train can serve as a privacy metrics, provided that key private attributes significantly contribute to the distance definition (see discussion in Limitations and Appendix C). The idea behind this viewpoint is that even when the model underfits, we still detect that synthetic data are actually closer to train than to test samples, in a meaningful statistical sense this can point to a risk of privacy leakage. Moreover, our controlled experiments with artificial leakage explicitly demonstrate that the proposed metric is sensitive to increasing levels of leakage: as the amount of copied information grows, the score consistently reflects this shift. In the image experiments, the “copycat” setup provides a concrete scenario where the generative model partially reproduces training samples, i.e., copying up to deformations, and our metric successfully identifies this behavior.
>
> > One potential improvement would be to employ a data generator with a controlled privacy parameter—for example, a differentially private generator with varying epsilon values—and evaluate whether the proposed privacy score correlates with these privacy levels.
>
> We thank the reviewer for this valuable suggestion. We agree that evaluating the metric on a generator with an explicit privacy parameter (e.g., a DP model with varying $\epsilon$) would provide an alternative validation. We will make every effort to include such an experiment during the rebuttal period.
>
> > Finally, it would be valuable to extend the analysis beyond membership inference attacks. Incorporating sensitive attribute inference or other privacy attack scenarios could help assess whether the proposed score also correlates with stronger leakage in more complex attack models.
>
> We thank the reviewer for this valuable suggestion. Extending the analysis to sensitive attribute inference and other attack models is indeed relevant for assessing broader forms of leakage, but we view this more as an important direction for future work, rather than being in the scope of the present one.

---

### Author Response · Authors · 2025-12-03

We would like to thank all reviewers for their valuable insights and aim to summarize main reviewer's concerns and outline the additions we included in the camera-ready version of the manuscript.

The main concerns raised were:
(i) whether PRIVET truly detects privacy risk rather than just overfitting or memorization and whether the proposed privacy score correlates with a controlled privacy parameter—for example, a differentially private generator with varying epsilon values.
(ii) the extent to which PRIVET is genuinely modality-agnostic and how it behaves across different embeddings and heterogeneous data types.
(iii) the clarity of our notion of interpretability and the empirical support for scalability and low-data robustness.
(iv) it was reproached that "PRIVET does not consistently outperform the In-Authenticity baseline" in the copycat experiment on image data.

To address (i), we added a new section on differential privacy, containing a differentially private training of a diffusion model on CIFAR10 and showing that PRIVET’s leakage scores vary monotonically with $\epsilon$ and recover a non-DP baseline (constructed in such a way that it overfits a subset of 1K samples) in weak-privacy regimes, thereby demonstrating that PRIVET is sensitive to controlled privacy levels rather than merely detecting overfitting or memorization.

To address (ii), we now emphasize that PRIVET is modality-agnostic at the level of its formulation, but requires a meaningful distance, which may in practice depend on choosing an appropriate representation. We provide concrete guidance: inspecting the 1-NN CDFs offers an immediate diagnostic of the underfitting/overfitting/leakage regime. We also clarified evidence for additional experiments showing that the NN-distance distribution follows the Extreme Value regime for heterogeneous tabular datasets (UCI Adult, UCI Bank; Appendix D).

To address (iii), we clarified our use of the term interpretable in the main text (in related work section, around Table.1). We also clarified evidence for behavior of competing metrics and ours in low dataset size regime, and for our scalability claims with a runtime comparison against other methods including PRIVET.

To address (iv), we moved the alternative PRIVET scores from the appendix to the main text and showed that, once the appropriate score for the underfitting-with-leakage regime is used, PRIVET remains sensitive across embeddings and image transforms, and consistently outperform the In-Authenticity baseline.

---

### Meta-Review · Area_Chair_eJ4b · 2026-01-06

**Summary:**

The paper studies a new privacy metric based on the nearest neighbour (NN) statistics, namely PRIVET. The privacy score is build upon the k-th smallest NN to the reference dataset, computed from samples. The proposed metric utilise the logratios and models like CDF of Weibull or Gumbel are used for extreme value properties. A low value indicates a statistical anomaly and may signal a potential privacy leak. Experiments are reported on CIFAR-10 via DINOv2 and wavelet embeddings. The paper argues the interpretability and scalability of PRIVET.

**Reviewer Concerns:**

- experiments on single RBM generated example;

 This has been addressed by promising additional experiment or pointed out by other reviewer on broad empirical coverage.

- sensitive representation and the notion of interpretable output

partially addressed and still unclear

- scalability concerns and privacy measure remains partially addressed

**Reviewer Scores:**

Part of the concerns are only addressed to a limited extent. Score may not change further.

---

### Decision · Program_Chairs · 2026-01-26

Reject